# Reanalysis of the 1761 transatlantic tsunami

Martin Wronna[1,3], Maria Ana Baptista[1,2], Jorge Miguel Miranda[1,3]

1 Instituto Dom Luiz, Faculdade de Ciências da Universidade de Lisboa, Portugal
Instituto Superior de Engenharia de Lisboa, Instituto Politécnico de Lisboa, Portugal
3 Instituto Português do Mar e da Atmosfera, IP, Lisboa, Portugal

*Correspondence to*: Martin Wronna (Mawronna@fc.ul.pt)

**Abstract.** The segment of the Africa-Eurasia plate boundary between the Gloria fault and the Strait of Gibraltar has been the setting of significant tsunamigenic earthquakes. However, their precise location and rupture mechanism remain poorly understood. The investigation of each event contributes to a better understanding of the structure of this diffuse plate boundary

and ultimately leads to a better evaluation of the seismic and tsunami hazard. The 31st March 1761 event is one of the few known transatlantic tsunamis. Macroseismic data and tsunami travel times were used in previous studies to assess its source area. However, no one discussed the geological source of this event. In this study, we present a reappraisal of tsunami data to show that the observations dataset is compatible with a geological source close to Coral Patch and Ampere seamounts. We constrain the rupture mechanism with plate kinematics and the tectonic setting of the area. This study favours the hypothesis

that the 1761 event occurred in the southwest of the likely location of the 1st November 1755 earthquake in a slow deforming compressive regime driven by the dextral transpressive collision between Africa and Eurasia.

## 1. Introduction

The coast along the southwest Iberian margin is prone to earthquakes and tsunamis. The earthquake and tsunami catalogues for the Iberian Peninsula and Morocco report three tsunamigenic earthquakes in the 18th century: 1722, 1755 and 1761

(Mezcua and Solares, 1983; Oliveira, 1986; Baptista and Miranda, 2009). While the 1722 event is believed to be a local event (Baptista et al., 2007), the 1st November 1755 and the 31st March 1761 earthquakes generated transatlantic tsunamis (Baptista et al., 1998a, b; Baptista et al., 2003; Baptista et al., 2006; Barkan et al., 2009). The source of the 1755 event has been extensively studied in recent years, e.g. Baptista et al. (1998a, b), Zitellini et al. (2001), Gutscher et al. (2006) and Barkan et al. (2009).

On the contrary, the tectonic source of 31st March 1761 remains poorly understood. The seismic catalogues present different earthquake locations: 10.00 W, 37.00 N (Mezcua and Solares, 1983) or 10.50 W, 36.00 N (Oliveira, 1986). Baptista et al. (2006), used macroseismic intensity data and tsunami travel time observations to locate the source circa 13.00 W, 34.50 N and estimated the magnitude in 8.5. The source location obtained by Baptista et al. (2006) places the 1761 event southwest of the South West Iberian Margin (SWIM) in the outer part of the Gulf of Cadiz (Fig. 1). The plate boundary between Eurasia and

Africa is not well defined in the SWIM area as the deformation is distributed over a large area. Here, a complex system of

faults accommodates the stress driven by the present-day tectonic regime that is constrained by NW-SE plate convergence between Africa and Eurasia at ~4 mm/yr (Argus et al., 1989; DeMets et al., 1994) and by the westward migration of the Cadiz Subduction slab ~2 mm/yr (Gutscher et al., 2012; Duarte et al., 2013).

The SWIM is dominated by large NE-SW trending structures limiting the Horseshoe Abyssal Plain (HAP) (Fig. 1). The NE-SW striking structures are the Coral Patch fault (CPF) (Martínez-Loriente et al., 2013), the Gorringe Bank fault (GBF) (Zitellini et al., 2009; Jiménez-Munt et al., 2010; Sallarès et al., 2013; Martínez-Loriente et al., 2014), the Horseshoe fault (HSF) (Gràcia et al., 2003; Zitellini et al., 2004; Martínez-Loriente et al., 2018) and the Marques de Pombal fault (MPF) (Gràcia et al., 2003; Terrinha et al., 2003; Zitellini et al., 2004) (Fig. 1). Other identified NE-SW trending structures are the Sao Vincente Fault (SVF) (Gràcia et al., 2003; Zitellini et al., 2004), the Horseshoe Abyssal Plain Thrust (HAT) (Martínez-Loriente et al., 2014), and to the south of the CPF the Seine Hills (SH) (Martínez-Loriente et al., 2013) (Fig. 1).

Large WNW-ESE trending dextral strike-slip faults (SWIM-Lineaments LN and LS) further characterise the SWIM cutting through the Gulf of Cadiz until the HAP (Zitellini et al., 2009; Terrinha et al., 2009, Rosas et al., 2009) (Fig. 1). To the south, the igneous Ampere and Coral Patch seamounts limit the HAP.

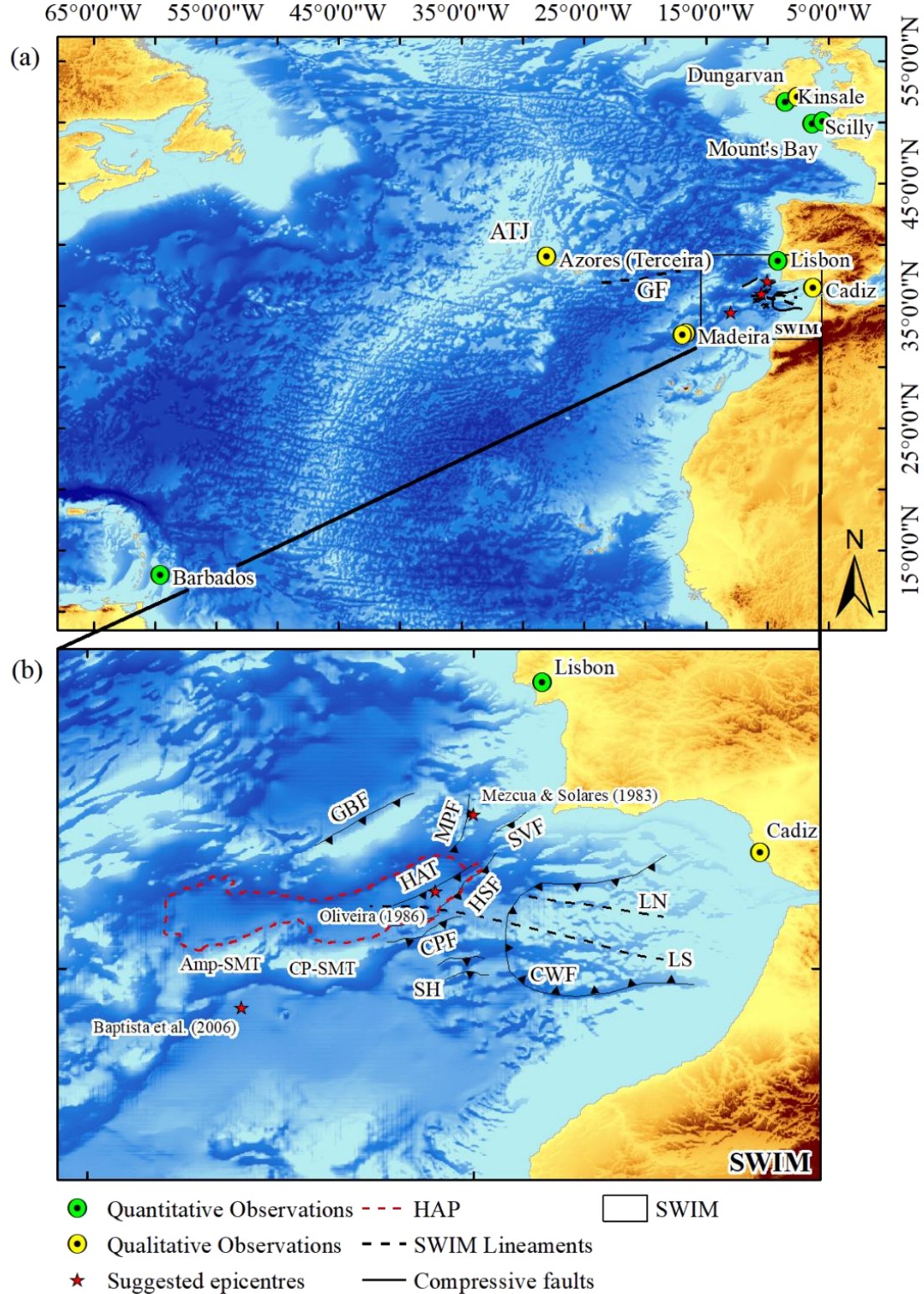

(a)

(b)

- ● Quantitative Observations
- ◉ Qualitative Observations
- ★ Suggested epicentres
- – – – HAP
- – – – SWIM Lineaments
- —— Compressive faults
- ☐ SWIM

**Figure 1. The red stars plot the source location by Oliveira (1986), Mezcua and Solares (1983) and Baptista et al. (2006). The green circles depict the quantitative tsunami observation points, and the yellow circles show the locations of the qualitative descriptions of the tsunami in 1761. The main features of the Azores Gibraltar fracture zone are the Azores Triple Junction (ATJ), the Gloria Fault (GF) and the Southwest Iberian Margin (SWIM). The inset shows the position of the Ampere seamount (Amp-SMT), the Coral Patch Seamount (CP-SMT) and the locations of the known faults. The black lines mark the faults, and the triangles indicate the direction of dip. The dashed black lines trace the main strike-slip faults. The known thrust faults are the Coral Patch Fault (CPF), the Cadiz Wedge Fault (CWF), the Gorringe Bank fault (GBF), the Horseshoe Fault (HSF) and the Marques de Pombal Fault (MPF). The shown strike-slip faults are the SWIM lineaments (LN) and (LS) and the Gloria Fault (GF). The dashed red line limits the Horseshoe Abyssal Plain (HAP).**

In this study, we investigate the geological source of the 1761 transatlantic tsunami. To do this, we start with a reappraisal of previous research, we analyse the tectonic setting of the area and propose a source compatible with plate kinematics. From this source, we compute the initial sea surface displacement. To propagate the tsunami, we build a bathymetric dataset based on GEBCO (2014) data to compute wave heights offshore the observations points presented in table 1. We also compute inundation using high-resolution digital elevations models comprising topography and bathymetry in Lisbon and Cadiz to compare the results with the observations. Finally, we use Cadiz and Lisbon observations in 1755 and 1761 to compare the size of the events.

## 2. Geodynamical context

The plate boundary between Africa and Eurasia in the NE Atlantic Ocean, the Azores – Gibraltar Fracture Zone (AGFZ), extends from the Azores Triple Junction (ATJ) to the Gibraltar Arc. The main features of the AGFZ are the ATJ; the Gloria Fault (GF) and the SWIM (Fig. 1). At the ATJ, active interplate deformation defines the plate boundary (Fernandes et al., 2006). The GF is a large W-E striking strike-slip fault with scarce seismicity (Laughton and Whitmarsh, 1974) with a strong Mw = 8.3 event on the 25th November 1941 (Gutenberg and Richter, 1949; Moreira, 1984; Baptista et al., 2016). The Gloria fault defines a sharp boundary between Eurasia and Africa (Laughton and Whitmarsh, 1974). Further East, towards the Gulf of Cadiz, in the plate boundary is not clearly defined (Torelli et al., 1997; Zitellini et al., 2009). Large-scale dynamics are imposed by convergence between Africa and Eurasia and by the westward propagation of the Gibraltar Arc. Most recent studies agree that the source of the 1755 Lisbon earthquake with a magnitude of about 8.5±0.3 is in the SWIM (Johnston, 1996; Baptista et al., 1998b; Zitellini et al., 1999; Gutscher et al., 2002; Solares and Arroyo, 2004; Ribeiro et al., 2006).

In the SWIM, two main sets of faults have been identified: Large NE-SW trending thrust faults and WNW-ESE trending dextral strike-slip faults.

Thrust faults include large NE-SW trending structures namely the Horseshoe Fault (HSF) (Gràcia et al., 2003; Zitellini et al., 2004; Terrinha et al., 2009; Martínez-Loriente et al., 2018), the Marquês de Pombal fault (MPF) (Gràcia et al., 2003; Terrinha et al., 2003; Zitellini et al., 2004), the Gorringe bank fault (GBF) (Zitellini et al., 2009; Jiménez-Munt et al., 2010; Sallarès et al., 2013; Martínez-Loriente et al., 2014) and the Coral Patch fault (CPF) (Martínez-Loriente et al., 2013) (Fig. 1). The GBF and the CPF bound the Horseshoe Abyssal Plain (HAP). The NE-SW striking thrusts are deep-rooted faults accompanied by morphological seafloor signatures. Moderate and small magnitude events (M<5) characterise the seismicity of the area. These

faults lie between the Gorringe Bank and the Strait of Gibraltar (Custódio et al., 2015). South of the HAP the Coral Patch ridge was identified to have a northern and a southern segment (Martínez-Loriente et al., 2013).

Other smaller NE-SW trending structures are the Sao Vincente Fault (SVF) (Gràcia et al., 2003; Zitellini et al., 2004), the Horseshoe Abyssal Plain Thrust (HAT) (Martínez-Loriente et al., 2014), and to the south of the CPF the Seine Hills (SH) (Martínez-Loriente et al., 2013) (Fig. 1).

The SWIM-Lineaments (LN and LS) (Fig. 1) have been interpreted as the present-day boundary between the Eurasia and Africa plates (Zitellini et al., 2009). They are large WNW-ESE trending dextral strike-slip faults with lengths of ~130 and 180 km for the LN and LS respectively. OBS monitoring captured numerous moderate-magnitude seismic events (Mw 3 – 5) at the intersection of the SWIM faults and NE-SW striking thrusts (Geissler et al., 2010; Silva et al., 2017). Ocean floor morphological signatures like en echelon folds and sets of undulations suggest the quaternary reactivation of the deep-rooted basement faults (Terrinha et al., 2009; Rosas et al., 2009). Terrinha et al. (2009) propose that the present-day deformation in the SWIM is accommodated by strain partitioning of dextral wrenching along the SWIM-Lineaments and thrusting along the NE-SW faults in the Gulf of Cadiz and the HAP. Bartolome et al., (2012) attributes the SWIM faults to have the capacity to trigger Mw > 8.0 earthquakes.

Considering the today known tectonic structures, the CPF is the largescale fault closest located to the source area of the 1761 event suggested by Baptista et al. (2006). This area located southwest of the SWIM faults is in a slow deforming compressive regime driven by the dextral transpressive collision between Africa and Eurasia. Hayward et al. (1999) showed the existence of widespread compressive structures in this region (Coral Patch and Ampere seamounts) based on shallow seismic reflection and side scan sonar data (Fig. 1 and 3). The tectonic deformation uplifted the oceanic crust showing the pervasive original NE-SW striking oceanic fabric formed during oceanic rifting (Hayward et al., 1999; Zitellini et al., 2009). The IGN seismic catalogues list a 6.2 magnitude around the Coral Patch on 11th of July 1915 (Instituto Geográfico Nacional, 2018).

Kinematic plate models (Argus et al., 1989; DeMets et al. 1999; Nocquet and Calais 2004; Fernandes et al., 2007) predict low convergence rates 3 - 5 mm per year between African plates and Eurasia. We used the global kinematic plate model Nuvel-1A. This model is a recalibrated version of the precursor model Nuvel-1 that implements rigid plates and data from plate boundaries such as spreading rates, transform fault azimuths, and earthquake slip vectors (DeMets et al., 1990). The NUVEL 1A model predicts a relatively conservative convergence rate of 3.8 mm per year in the area close to the source area determined by Baptista et al. (2006) for the 1761 tsunami (Fig. 2).

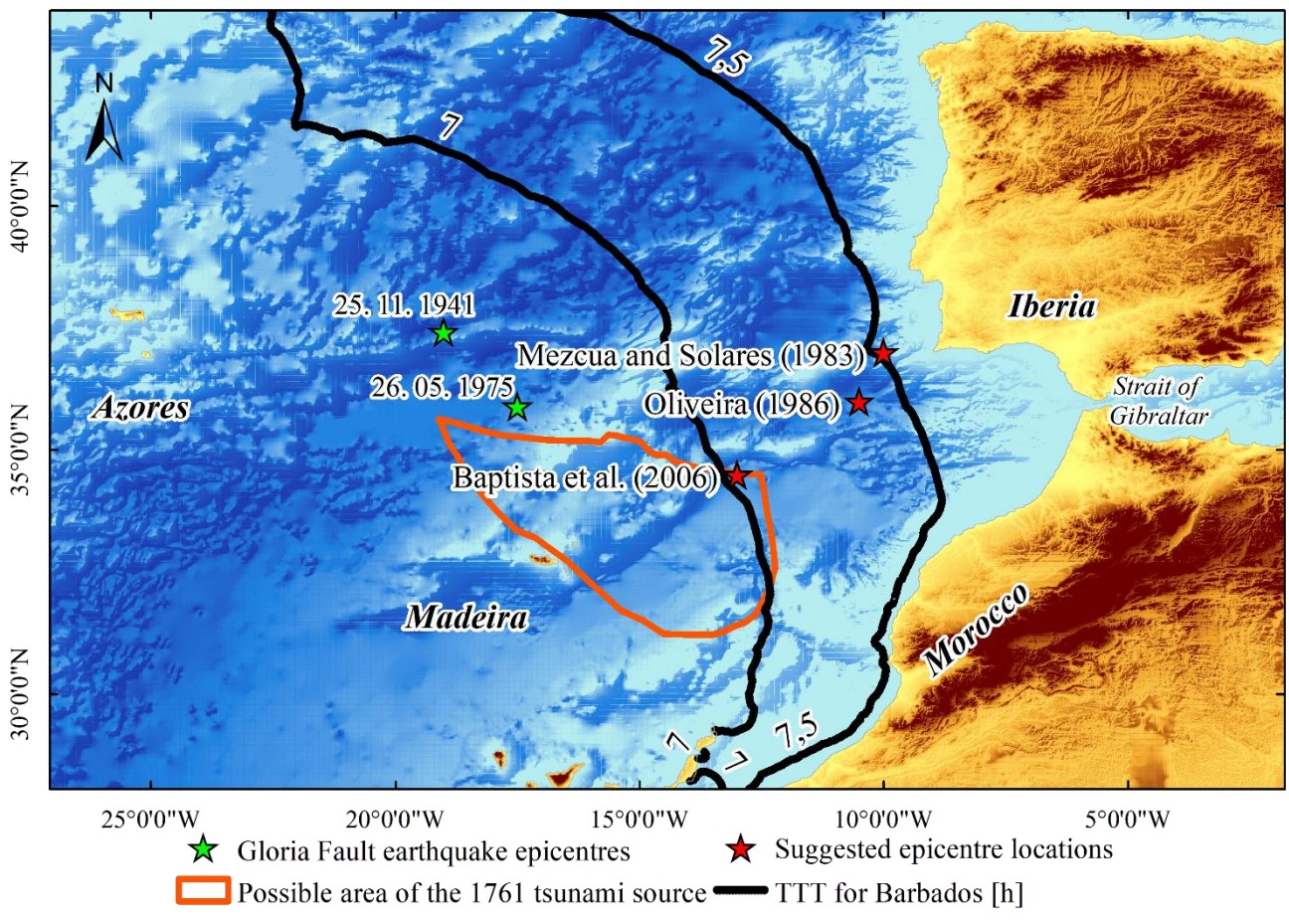

**Figure 2. The red stars show the proposed source locations for the 1761 earthquake. The green stars present the epicentres of the two strong magnitude events in the Gloria Fault zone, and the black lines show the backward ray tracing contours for the tsunami travel time (TTT) of 7 – 7.5 hours to Barbados. The limited orange area defines the results obtained using macroseismic analysis combined with backward ray tracing but discarding the TTT for Barbados by Baptista et al. (2006).**

Consequently, we propose a fault extending from the western segment of the CPF towards the epicentre proposed by Baptista et al. (2006). We draw the circle around the Euler pole at -20.61 W, 21.03 N according to the plate kinematic model Nuvel 1-A using Mirone suite (Luis, 2007). To do this, we choose Africa as the fixed plate and Eurasia as the moving plate and draw the circle at the centre of the fault in figure 3. We compute the convergence rate (3.8 mm per year) and plot the tangent velocity vector along the circle (Fig. 3). For this fault, we test different earthquake fault parameters (table 2) and compute the co-seismic deformation using the Mansinha and Smiley equations (Mansinha and Smiley, 1971) implemented in Mirone suite (Luis, 2007). We assume that the initial sea surface elevation mimics the sea bottom deformation and we use it to initiate the tsunami propagation model.

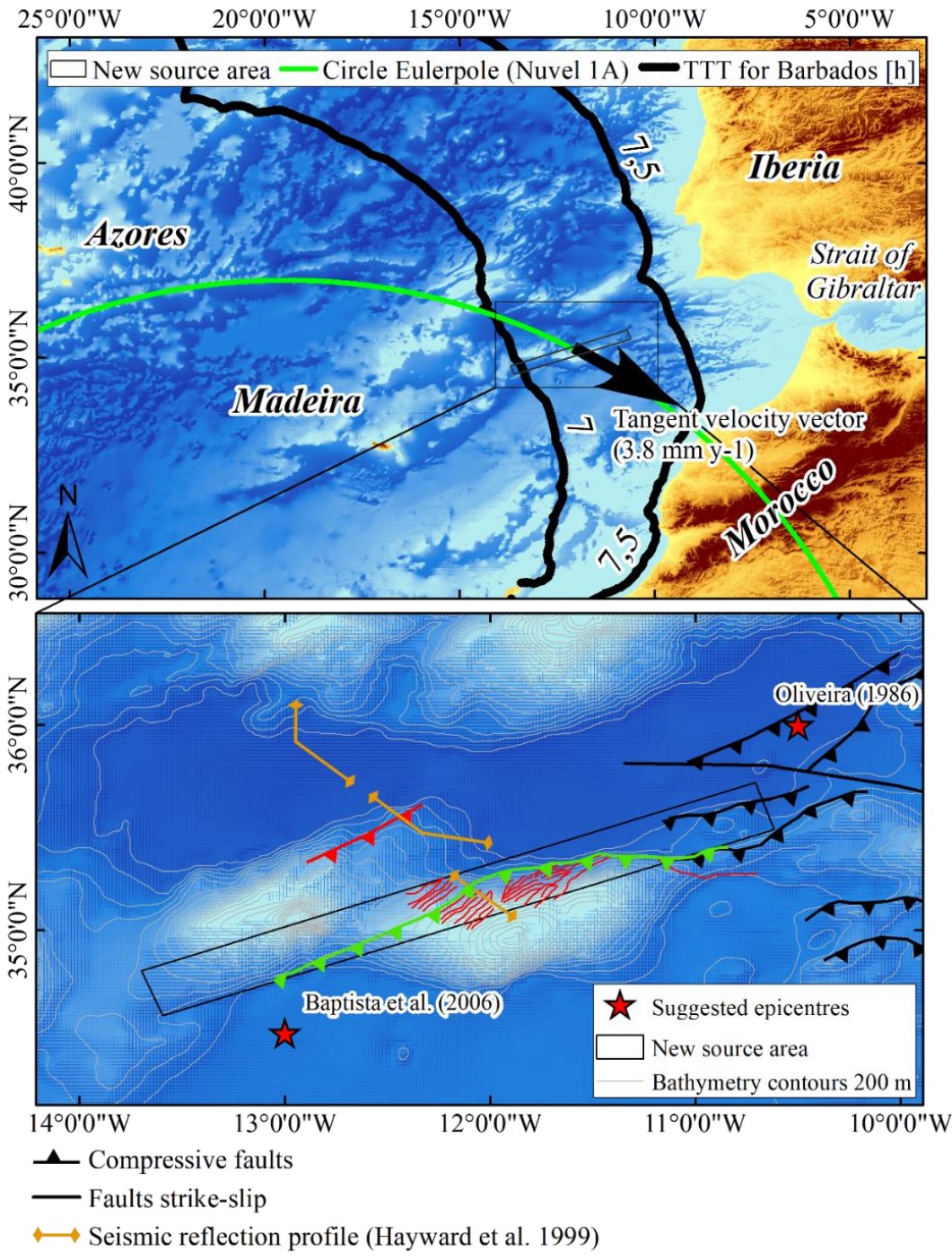

**Legend:**

▲— Compressive faults

—— Faults strike-slip

◆—◆ Seismic reflection profile (Hayward et al. 1999)

▲— Compressive fault (Hayward et al. 1999)

—— Compressive structures (Hayward et al. 1999, Zitellini et al. 2009)

▲— Proposed fault in this study

**Figure 3.** Circle around the Euler pole at the proposed possible source location. The model Nuvel 1A (DeMets et al., 1994, 1999) computes a 3.8 mm y-1 convergence. We plot the tangent velocity vector close to the candidate source. The black lines depict the backward ray tracing contours in hours, for a tsunami travel time (TTT) of 7-7.5 hours to Barbados. In the inset, the black lines plot the thrust and strike-slip faults. The red lines depict the faults and structures proposed in Hayward et al. (1999). The orange lines show the location of the seismic profiles for the areas of the Coral Patch and Ampere seamount (Hayward et al., 1999). The green line identifies the proposed tectonic structure used in this study. The red stars are the locations of the closest proposed epicentres (Baptista et al., 2006 and Oliveira, 1986).

## 3. Reassessment of historical data on the 1761 tsunami

Baptista et al. (2006) and Baptista and Miranda (2009) present most of the tsunami observations used herein. Here, we focus on the observations of wave heights, periods, inundation and duration of the sea disturbance that we summarise in table 1. We only reassess the observations in Barbados and Cadiz.

Barbados: Baptista et al. (2006) discarded the arrival time observation in Barbados. However, we find that this is compatible with the source location. The observations report the tsunami arrival at 4 p.m. local time (Mason, 1761; Annual Register, 1761; Borlase, 1762). If we use a solar time difference between Lisbon and Barbados of 3.5 h, as in Baptista et al. (1998a, b), we conclude for a tsunami travel time of 7-7.5h. To validate this TTT, we did a backward ray tracing simulation with a point source in Barbados (see Fig. 2 and 3) and we find that the TTT is compatible with the source area.

Cadiz: The Journal des Matiéres du Temps (Journal Historique, 1773) describes the occurrence of an earthquake in April 1773 and compares it with the 31st March 1761 event. The report concludes that no tsunami was observed in 1773 and suggests a withdraw of the sea after the 31st March 1761 earthquake in the city; however, there are no accounts of inundation neither for the city nor the causeway. We include this observation to constrain the proposed source better.

Table 1 presents a summary of all historical data relevant to the tsunami simulation. Figure 1 shows the locations of the tsunami observations. Wave heights always refer to the maximum positive amplitude above the still water level.

**Table 1. Summary of the available data of the 1761 tsunami at the time. The column TTT lists the observed Tsunami Travel Times. The column polarity indicates the first movement of the sea upward (U) or downward (D).**

| Location | Lon. [°] | Lat. [°] | Local Time | TTT [h] | Wave height [m] | Polarity | Period [min] | Duration | Source |
|---|---|---|---|---|---|---|---|---|---|
| Lisbon | -9.13 | 38.72 | 13:15 | 1.25 | 1.2 - 1.8 | - | 6 | Lasted until night | Unknown (1761); Molloy (1761); Borlase (1762) |
| Cadiz | -6.29 | 36.52 | - | - | - | D | - | - | Journal des Matieres du Temps (1773) |
| Kinsale | -8.51 | 51.67 | 18:00 | 6 | 0.6 | U | 4 | Repeated several times | Annual Register (1761); Borlase (1762) |
| Scilly Islands | -6.38 | 49.92 | 17:00 | 5 | 0.6 - 1.2 | U | - | > 2 hours | Borlase (1762) |
| Mount's Bay | -5.48 | 50.08 | 17:00 | 5 | 1.2 - 1.8 | U | 12 | 1 hour | Borlase (1762) |
| Dungarvan | -7.48 | 51.95 | 16:00 | 4 | - | - | - | 5 hours | Borlase (1762) |
| Barbados | -59.57 | 13.03 | 16:00 | 7 - 8 | 0.45 - 0.6 | - | 8 | 4 hours but lasted until 6 in the morning. | Mason (1761); Annual Register (1761) |
| | -59.57 | 13.03 | | | 0.6 | - | 3 - 6 | Increased again at ten for short time then decreased. | Borlase (1762) |
| Madeira | -16.91 | 32.62 | - | - | ~1; higher in the East | - | - | Lasted longer in the East than in the South. | Heberden (1761) |
| Azores | -27.22 | 38.65 | - | - | Large | U | Some min. | 3 hours | Fearns (1761) |

## 4. Tsunami Simulations

### 4.1 The numerical model

We used the code NSWING (Non-linear Shallow Water model wIth Nested Grids) for numerical tsunami modelling. The code solves linear and non-linear shallow water equations (SWEs) in a Cartesian or spherical reference frame using a system of nested grids and a moving boundary condition to track the shoreline motion based on COMCOT (Cornell Multi-grid Coupled Tsunami Model; Liu et al., 1995; 1998). The code was benchmarked with the analytical tests presented by Synolakis et al. (2008) and tested in Miranda et al. (2014) and Baptista et al. (2016), Wronna et al. (2015) and Omira et al. (2015).

For Cadiz and Lisbon only, where high-resolution bathymetric data was available, we employed a set of coupled nested grids with a final resolution of 25 m to compute inundation. We compute a new bathymetric dataset using the nautical charts close to the coast or LIDAR data to build a Digital Terrain Model to compute inundation in Lisbon and Cadiz. Close to the tsunami source we interpolate bathymetry data (GEBCO, 2014) to obtain a 1600 m grid cell size. We apply a refinement factor of 4 for the four nested grids. Consequently, the intermediate grids have a resolution of 100 m and 400 m respectively. In Cadiz, we use the soundings and coastline of historical nautical charts from the 18[th] century (Bellin, 1762 and Rocque, 1762) to compute a Paleo Digital Elevation Model (PDEM) (Wronna et al., 2017). To do this, we geo-referenced the old nautical charts and use the modern-day DEM (UG-ICN, 2009) to implement the information from the historical charts. According to Wronna et al. (2017), we systematically remodelled bathymetry and the coastline.

To initiate the tsunami propagation model, we compute the co-seismic deformation according to the half-space elastic theory (Mansinha and Smylie, 1971) implemented in Mirone suite (Luis, 2007). Assuming that water is an incompressible fluid we translate the sea bottom deformation to the initial sea surface deformation and set the velocity field to zero for the time instant $t = 0$ s. We run the model for 10-hour propagation time to ensure that the tsunami reaches all observation points.

We compute the offshore wave heights for points located close to the observation points (Fig. 1) using Virtual Tide Gauges (VTG). We include the coordinates and depths of the VTG in the tables 3 and 4 in section 5. For transatlantic propagation, we consider the Coriolis effect in the tsunami simulation. We checked all tsunami simulations against historical data.

For the locations in Ireland, the United Kingdom, the Azores, Madeira and Barbados, we estimate the wave heights near the shore using the Green's Law (Green, 1838), following Hebert and Schindelé (2015) and Davies et al. (2017). Hebert and Schindelé (2015) concluded that the extrapolation for depths between 10 and 1 m generally allowed for a good fit with the observations for the 2004 Indian ocean tsunami. The Green's Law is based on the linear shallow water wave equations and allows to quickly approximate the amplification of wave heights at a shallower depth close to the shore when considering a plane beach. The wave height increases to the fourth root of the ratio between the depth at the shore and the water depth at the VTG. We extrapolate the maximum wave height values between the depths of the VTG (table 3 and 4) to points located at 5 m depth.

$$h_s = \sqrt[4]{\frac{d_s}{d_d}} * h_d \qquad\qquad \text{Eq. (1)}$$

Where $h_s$ and $h_d$ are the wave heights at the shore and the VTG respectively, and $d_s$ and $d_d$ are the depths at the shore and the VTG respectively. We use a constant value of 5 m which is sufficiently close to the shore to be observed by eyewitnesses. The results of the approximation according to the Greens Law are presented in table 3 and 4.

### 4.2 Testing the hypothesis

In the 20[th] century, two strong magnitude earthquakes occurred in the Gloria Fault (GF) area. Because of this, we tested the compatibility of the tsunami observations in 1761 with the tsunamis produced by the earthquakes of the 25[th] November 1941 (Lynnes and Ruff, 1985; Baptista et al., 2016) and 26[th] May 1975 (Kaabouben et al., 2009). We use the fault plane parameters and rupture mechanism presented in Baptista et al. (2016) and Kaabouben et al. (2008) for the 1941 and 1975 events respectively. The fault dimensions and slip were made compatible with an 8.5 magnitude event using the scaling laws proposed
by Wells and Coppersmith (1994), Manighetti et al. (2007), Blaser et al. (2010) and Matias et al. (2013).
These two events produce less than one-meter wave height in the North East Atlantic and were barely observed wave in the Caribbean Islands (Baptista et al., 2016; 2017). Moreover, the epicentres of the 25th November 1941 and 26th May 1975 are located outside the area determined by Baptista et al. (2006). As expected, the TTTs do not agree with those reported in 1761; therefore, we excluded the GF as a candidate source for the 1761 event and do not consider their results for discussion.
The candidate fault area is centred at 12.00 W, 35.00 N to the west of the large NE/SW striking compressive structures (Martínez-Loriente et al., 2013) and 85 km northeast of the epicentre suggested by Baptista et al. (2006) (Fig. 3). We considered the fact that the historical accounts indicate an earthquake and tsunami less violent than in 1755. To account for this, we used the fault dimensions presented in table 2 corresponding to a magnitude 8.4-8.5 earthquake (Baptista et al., 2006); consequently, the wave heights in Lisbon and Cadiz are smaller than those observed in the 1755 tsunami (Baptista et al., 1998). The fault
dimensions (length and width) presented in table 2 are compatible with the scaling laws of Wells and Coppersmith (1994), Manighetti et al. (2007), Blaser et al. (2010) and Matias et al. (2013).
**Hypotheses A and A-MS:** Here we use a strike angle compatible with the study by Martínez-Loriente et al., (2013) that follows the morphology of the Coral Patch scarp and seamount (Fig. 1 and Fig. 3). To take into account the tectonic regime of the source area we choose fault plane parameters compatible with a structure of compressive nature. The velocity vector
predicted by NUVEL 1A (Fig. 3) together with the short tsunami wave periods (4-12 minutes) reported in 1761 (table 1) are in line with the chosen dip angle of 40 degrees (table 2). On the other hand, Martínez-Loriente et al. (2013) suggest for the Coral Patch Faults dip angles of 30±5 degrees dip and a rake angle of 90 degrees. These authors also conclude that the fault root is between 7 and 13 km depth. We approximate the rake angle according to the difference between the convergence arrow given by the circle around the Euler Pole and the fault plane (Fig. 3).
The wave period in Lisbon produced by this candidate source is 30 minutes. This value is not compatible with the observations (Table 1). Trying to solve this problem, we implemented a multi-segment fault here called A-MS. This multi-segment solution consists of four segments each 50 km. The four segments are placed adjacent to each other, and the rupture mechanism is equal for each segment as in hypothesis A with a mean slip of 11 m (Table 2). To investigate the slip distribution, we tested three

setups: (1) maximum slip towards the SW, (2) maximum slip towards the NE and (3) maximum slip close to the centre of the fault. In the first setup the withdraw historical observed at Cadiz is less evident and produces little inundation in Lisbon; in the second setup there is inundation at Cadiz, which is not supported by historical data. All these results led us to select the maximum slip at the centre of the fault. The slip of each segment is presented in table 2. The synthetic waveforms are presented
in figure 5 and discussed in sections 5 and 6.

**Hypothesis B:** Finally, we test an alternative hypothesis here called B with a larger strike-slip component compared to hypothesis A. This also results in larger fault length and a steeper dip angle. Here, we consider a rupture along a fault plane rotated about 180° when compared to hypothesis A. To do this, we selected compatible strike and rake angles that result in a sinistral inverse lateral rupture (table 2). The implementation of the different setups of slip distribution in solution B does not
improve the quality of the results, therefore we only consider a single segment fault for this hypothesis. The synthetic waveforms are presented in figure 7 and discussed in sections 5 and 6.

**Table 2. The fault dimensions and parameters used herein to investigate candidate sources of 1761 event. We describe hypotheses (Hyp.) A-MS, A and B by the fault parameters length (L), width (W), strike, dip, rake, slip and depth. The slip values for hypothesis A-MS are listed for each segment from west to east. Additionally, we present the moment magnitude (Mag.), the assumed shear**
**modulus (μ) and the focal mechanism.**

| Scenario | L [km] | W [km] | Strike [°] | Dip [°] | Rake [°] | Slip [m] | Depth [km] | Mag. | μ [Pa] | Focal mechanism |
|---|---|---|---|---|---|---|---|---|---|---|
| Hyp. A-MS | 4 x 50 | 50 | 76 | 40 | 135 | 7/15/15/8 | 10 | 8.4 | $4*10^{10}$ | |
| Hyp. A | 200 | 50 | 76 | 40 | 135 | 11 | 10 | 8.4 | $4*10^{10}$ | |
| Hyp. B | 280 | 50 | 254.5 | 70 | 45 | 15 | 10 | 8.5 | $4*10^{10}$ | |

## 5. Results

We present the results of hypothesis A-MS and B. Hypothesis A-MS has a more significant inverse component compared to hypothesis B. Once the results of hypotheses A and A-MS produce same wave height values, but the latter produces shorter periods, we opt to present the results for hypothesis A-MS. Figures 4-7 show the maximum wave height and the synthetic
tsunami at the virtual tide gauges (VTG) computed offshore of each observation point of hypothesis A-MS and B. Tables 3 and 4 summarise these results. The wave height, as mentioned in section 3, represents the maximum positive amplitude above the still water level, which is set to be 0 in the tsunami simulation. Table 3 and 4 present the geographical coordinates and depths of the VTGs. To compare the synthetic wave heights with the observations for the locations in Mount's Bay, Scilly

Islands, Kinsale, Dungarvan, Azores, Madeira and Barbados we used the Green's Law (Green, 1838) to extrapolate the wave height values for the maximum wave between the depths of the VTG to points located at 5 m depth. For Lisbon and Cadiz, where high-resolution bathymetry is available we used two sets of nested grids and computed the tsunami inundation. Here the VTGs are located close to the shore, and the application of the Green's Law is not necessary.

## 5.1 Hypothesis A-MS

Figures 4 and 5 show the distribution of the maximum wave height and the respective synthetic tsunami records for hypothesis A-MS.

Analysis of figure 4 shows wave heights exceeding 4 m in the Gulf of Cadiz. At some points along the coast of Morocco maximum wave heights are about 5 m. In Great Britain, at the Scilly Islands and Mount´s Bay maximum wave heights vary between 1.7 and 1.9 m. Along the south coast of Ireland, in Kinsale and Dungarvan the tsunami simulation predicts a 1 m maximum wave height. At the eastern coast of Madeira Island, the wave heights reach 1 m whereas on the southern part of the island the wave heights are smaller. At the Azores close to Terceira Island wave heights are slightly higher than 2.5 m along the south coast of the island. The wave heights in the south of Barbados reach 0.5 m.

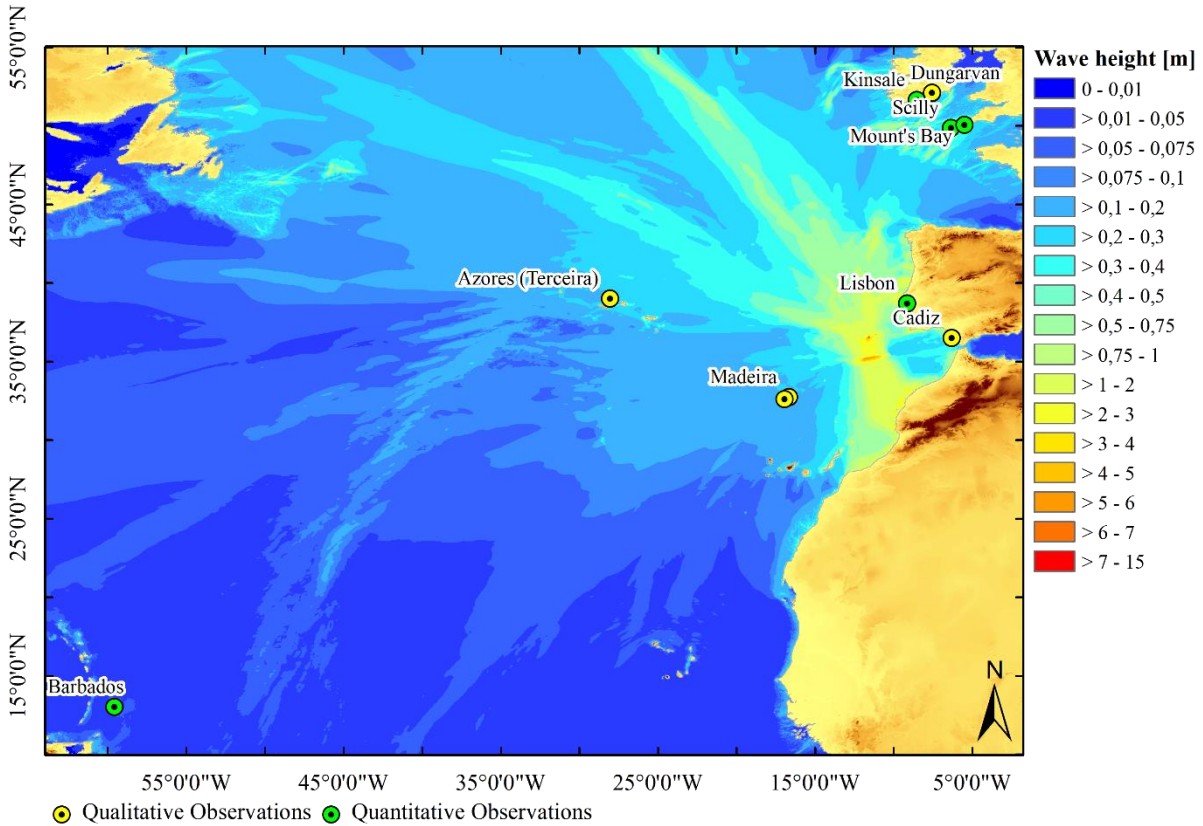

**Figure 4. Maximum wave height distribution (colour scale in m) in the Atlantic basin produced by the source of hypothesis A-MS.**

In Lisbon, the synthetic waveform shows a first peak of 1.4 m with a maximum value close to 1.8 m for the third wave, after two hours and twenty minutes of tsunami propagation. The TTT to Lisbon is 1 hour and 10 minutes and the first wave has a period of 20–25 minutes (Table 3 and Fig. 5 (a)). In Cadiz, the synthetic tsunami waveform shows a drawdown 1 hour after
5  the earthquake with a negative amplitude of 0.6 m and a maximum wave height of 2.4 m (table 3 and Fig. 5 (a)).

The Scilly Islands synthetic tsunami waveform shows a TTT of 4 hours and a maximum peak exceeding 0.4 m with 15 minutes period. In Mount´s Bay, TTT is 4 hours and 30 minutes and the maximum wave height is 0.5 m with 15 minutes period. In Kinsale, the tsunami model computes a TTT of 4 hours and 15 minutes. The maximum wave height there is about 0.5 m with a period shorter than 15 minutes. In Dungarvan, the tsunami arrives 5 hours after the earthquake. All VTGs in northern Europe
10  recorded the first wave as leading elevation wave (Fig. 5 (b and c)).

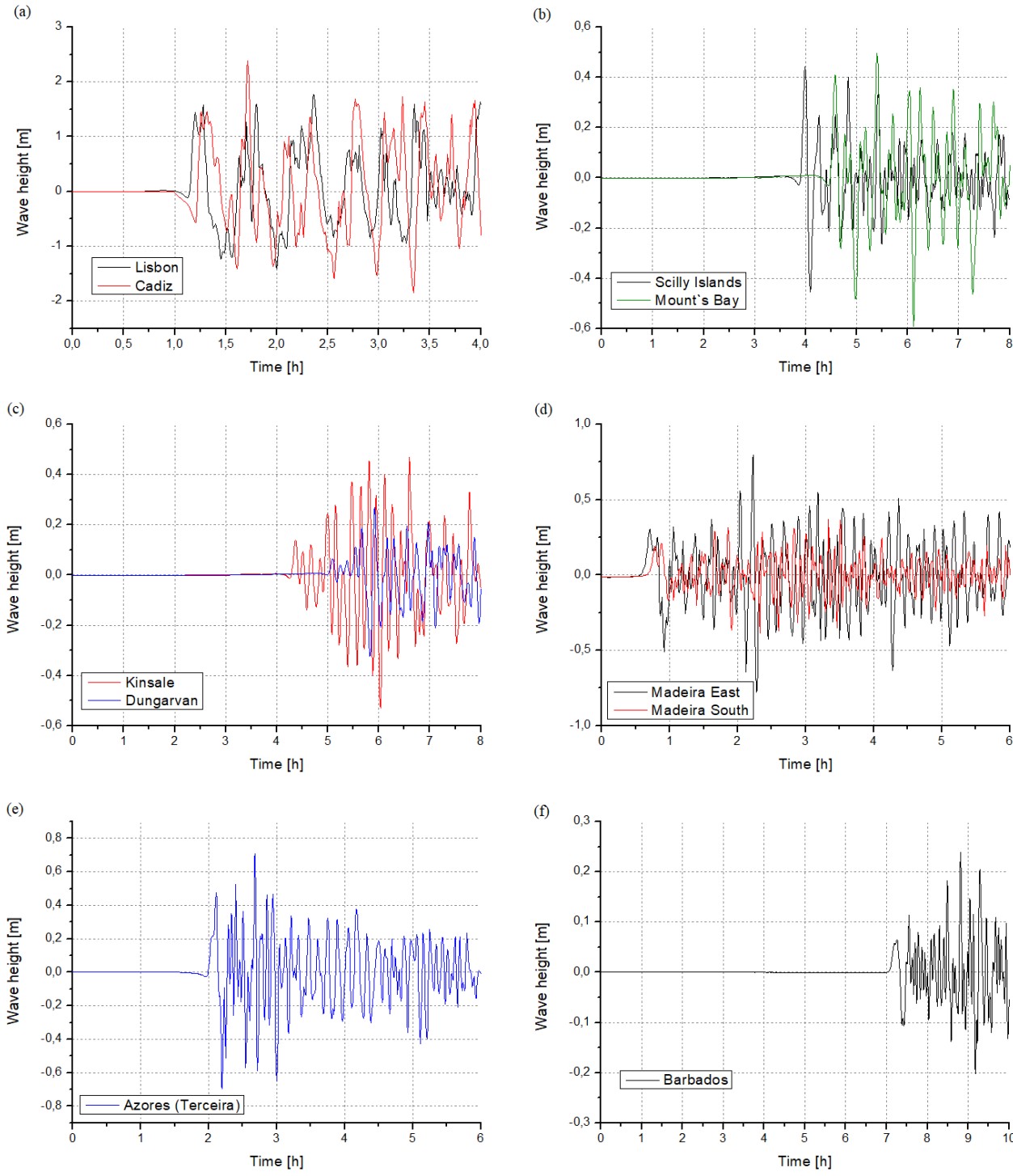

**Figure 5. VTG records for hypothesis A-MS at the coordinates of the locations presented in table 3.**

In Madeira, hypothesis A-MS produces maximum wave heights at the VTG of 0.8 m in the eastern part of the island and about 0.4 m, in the southern part; the TTT to the east and southern coast of the island is half an hour and 40 min respectively (Fig. 5 (d)). In the Azores, close to the island of Terceira, the wave heights reach approximately 0.7m (Fig. 5 (e)).

In Barbados, hypothesis A-MS produces the first wave of about 0.1 m after about 7 hours with about 30 minutes period. Only after 9 hours and 30 minutes, the wave height exceeds 0.2 m (Fig. 5(f)).

We applied the Green's Law in all locations except Lisbon and Cadiz to extrapolate the maximum wave height values to a depth of 5 m close to the shore to compare the values with the observations in section 3. We present the maximum wave height values after application of Green's Law in table 3.

**Table 3. Results of the VTGs for hypothesis A-MS. The column TTT lists the observed Tsunami Travel Times. The column polarity indicates the first movement of the sea upward (U) or downward (D).**

| Local | | VTG coordinates & depth | | | TTT | Wave height [m] | | | | Polarity | Period |
|---|---|---|---|---|---|---|---|---|---|---|---|
| | | Lon. [°] | Lat. [°] | d [m] | | First | max. | Green's Law | Obs. | | |
| Lisbon | | -9.136 | 38.706 | 3 | ~ 1 h 10 min | 1.6 m | 1.8 m | nesting | 1.2 – 1.8 m | D | < 30 min |
| Cadiz | | -6.291 | 36.524 | 4 | ~ 1 h | -0.6 m | 2.4 m | nesting | - | D | ~ 30 min |
| Scilly Islands | | -06.383 | 49.85 | 50 | ~ 4 h | 0.4 m | 0.4 m | 0.7 m | 0.6 – 1.2 m | U | ~ 15 min |
| Mount´s Bay | | -05.48 | 50.08 | 26 | ~ 4 h 30 min | 0.4 m | 0.5 m | 0.8 m | 1.2 – 1.8 m | U | ~ 15 min |
| Kinsale | | -08.500 | 51.653 | 28 | ~ 4 h 15 min | 0.1 m | 0.5 m | 0.8 m | 0.6 m | U | < 15 min |
| Dungarvan | | -07.479 | 51.949 | 50 | ~ 5 h | 0.1 m | 0.3 m | 0.5 m | - | U | < 15 min |
| Madeira | E | -16.666 | 32.750 | 51 | ~ 30 min | 0.3 m | 0.8 m | 1.4 m | - | U | ~ 30 min |
| | S | -16.926 | 32.619 | 51 | ~ 40 min | 0.2 m | 0.4 m | 0.7 m | - | U | ~ 30 min |
| Azores | | -27.150 | 38.800 | 53 | ~ 2 h | 0.5 m | 0.7 m | 1.3 m | - | U | ~ 15 min |
| Barbados | | -59.566 | 13.033 | 50 | ~ 7 h | 0.1 m | 0.2 m | 0.4 m | 0.45 – 0.6 m | U | ~ 30 min |

## 5.2 Hypothesis B

In hypothesis B the dip angle was increased relative to hypothesis A resulting in the dominant strike-slip mechanism. In figure 6, we depict the maximum wave height for option B.

Analysing figure 6 we find maximum wave heights of 15 m along the coast of Morocco. In the Gulf of Cadiz, the wave heights do not exceed 2 m. In Great Britain, at the Scilly Islands the maximum wave height is close to 2.3 m, and in Mount´s Bay, the maximum wave height values reach 1.8 m. For the locations in Ireland, Kinsale and Dungarvan, the maximum wave heights exceed 1.4 m. The eastern part of Madeira experiences wave heights greater than 2.5 m, decreasing towards the southern parts of the Island (Fig. 6). The maximum wave height exceeds 5.5 m on the eastern side of the island of Terceira in the Azores. For Barbados, this source computes maximum wave heights exceeding 0.7 m.

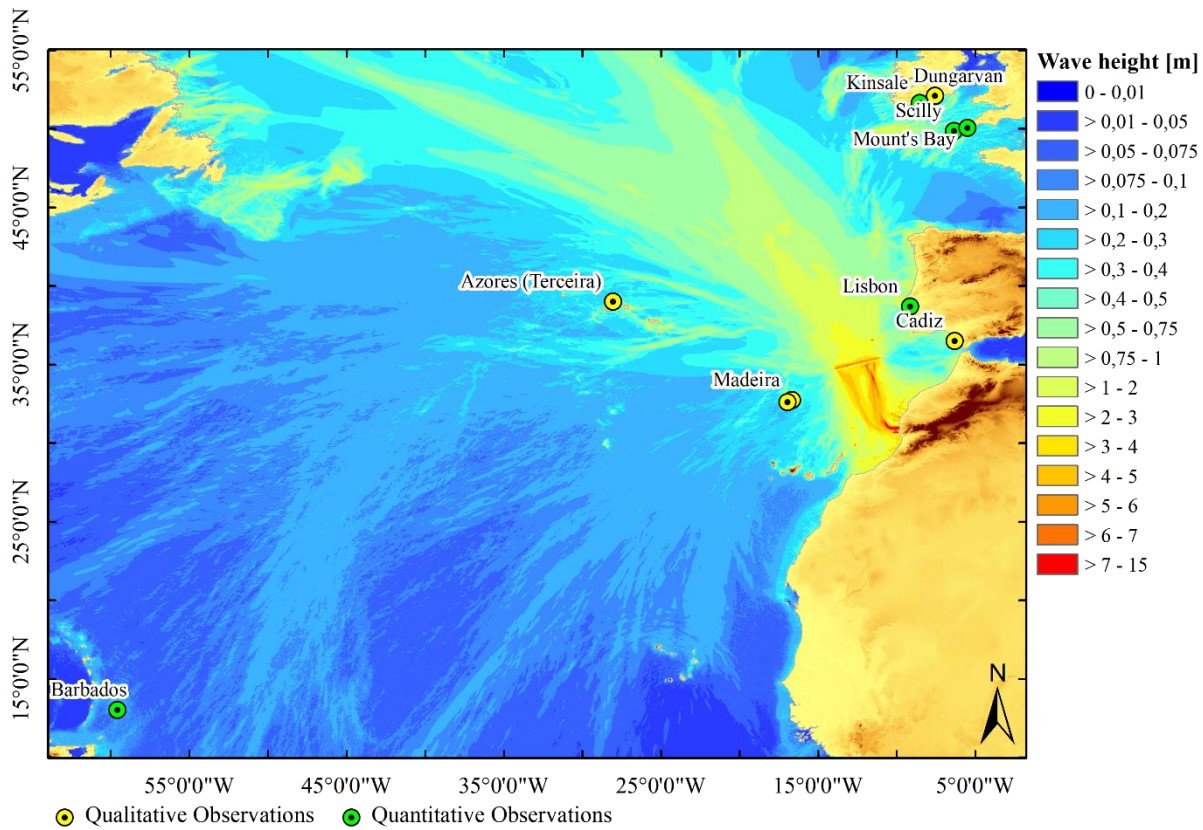

**Figure 6. Maximum wave height distribution (colour scale in m) in the Atlantic basin produced by the source of hypothesis B.**

Figure 7 presents the corresponding synthetic tsunami waveforms at the VTGs. Table 4 gives a summary of the results. The analysis of the synthetic waveforms shows that a small withdraw of about 0.2 m arrives in Lisbon after 1 hour and 15 minutes followed by a water surface elevation of 0.9 m. The third wave has a maximum positive amplitude of 2.2 m (Fig. 7 (a)).

The maximum wave heights at the Scilly Islands is 0.5 m (Fig. 7 (b)). The first wave reaches 0.4 m, arriving close to 4 h after the earthquake. The synthetic tsunami waveform shows around 15-minute wave period. In Mount`s Bay, the first wave of 0.4 m arrives after 4 hours and 30 minutes with a 15-minute wave period (Fig. 7 (b)). Here, the maximum wave height, 0.7 m, comes more than 6 hours after the earthquake. In Kinsale, hypothesis B produces a maximum wave height of 0.6 m. The first wave of 0.2 m wave height in the VTG arrives after 4 hours and 15 minutes of tsunami propagation; here, the period is shorter than 15 min (Fig. 7 (c)).

In Madeira, the first and the maximum wave heights are greater in the eastern part of the island compared to the southern part. Maximum wave heights values reach 1.4 m in the east part of Madeira and 1.1 m in the south part of Madeira (Fig. 7 (d)). In the Azores, the wave height for Terceira island reaches up to 2.4 m (Fig. 7 (e)).

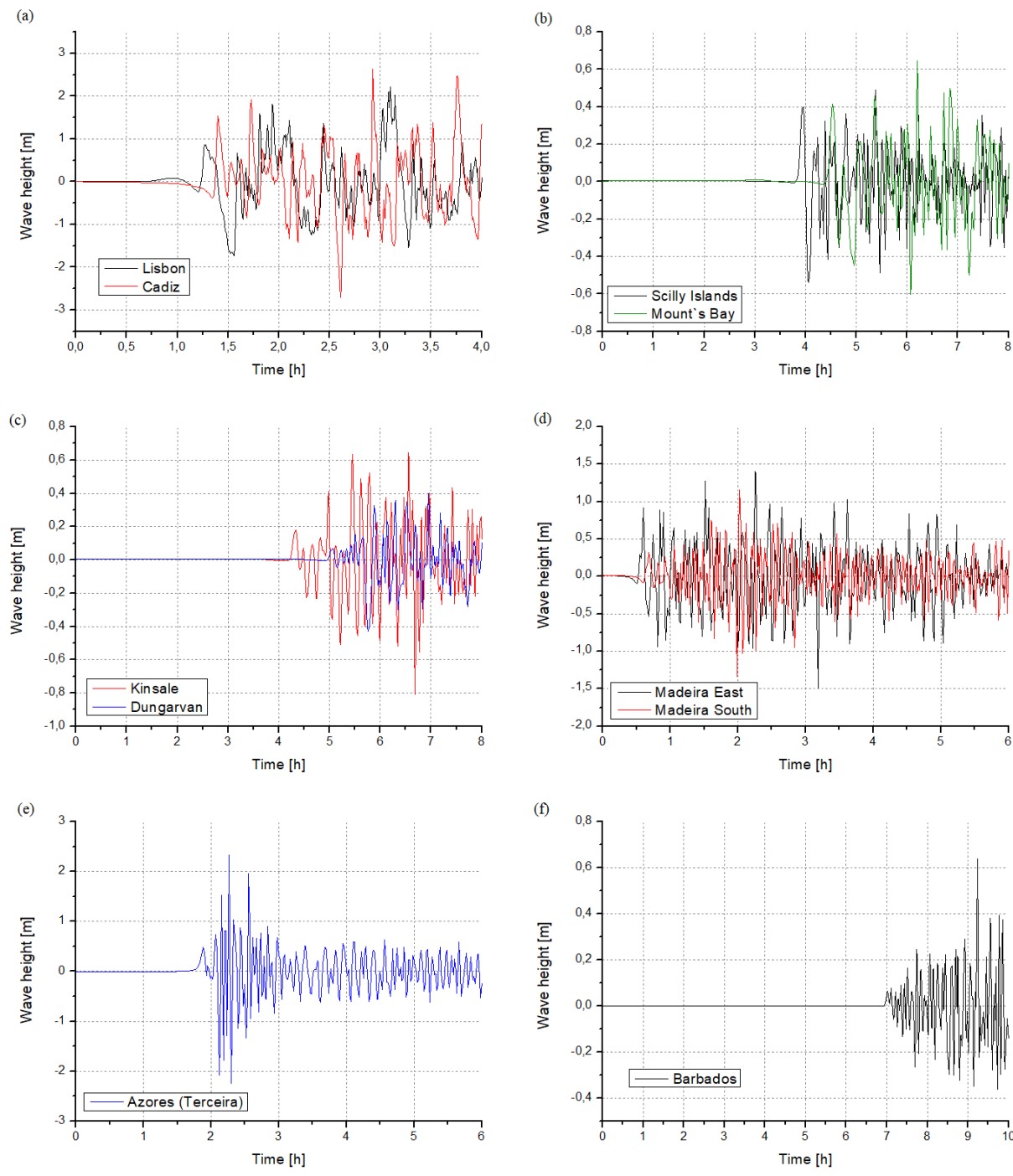

**Figure 7. VTG records for hypothesis B at the coordinates of the locations presented in table 4.**

Hypothesis B predicts a tsunami travel time of 7 hours to Barbados with the first peak of less than 0.1m and a maximum peak of 0.6 m after 9 hours and 15 minutes (Fig. 7 (f)). The first wave has a period slightly below 15 minutes. Table 4 gives a summary of the results for hypothesis B.

We also applied the Green's Law for this solution. We present the maximum wave height values after application of Green's
Law in table 4.

**Table 4. Results of the VTGs for hypothesis B. The column TTT lists the observed Tsunami Travel Times. The column polarity indicates the first movement of the sea upward (U) or downward (D).**

| Local | | VTG coordinates & depth | | | TTT | Wave height [m] | | | | Polarity | Period |
|---|---|---|---|---|---|---|---|---|---|---|---|
| | | Lon. [°] | Lat. [°] | d [m] | | First | max. | Green's Law | Obs. | | |
| Lisbon | | -9.136 | 38.706 | 3 | ~ 1 h 15 min | 0.9 m | 2.2 m | nesting | 1.2 – 1.8 m | D | > 30 min |
| Cadiz | | -6.291 | 36.524 | 4 | ~ 1 h | -0.4 m | 2.6 m | nesting | - | D | ~ 30 min |
| Scilly Islands | | -06.383 | 49.85 | 50 | < 4 h min | 0.4 m | 0.5 m | 0.9 m | 0.6 – 1.2 m | U | ~ 15 min |
| Mount´s Bay | | -05.48 | 50.08 | 26 | ~ 4 h 30 min | 0.4 m | 0.7 m | 1 m | 1.2 – 1.8 m | U | ~ 15 min |
| Kinsale | | -08.500 | 51.653 | 28 | ~ 4 h 15 min | 0.2 m | 0.6 m | 1 m | 0.6 m | U | < 15 min |
| Dungarvan | | -07.479 | 51.949 | 50 | ~ 5 h | 0.1 m | 0.4 m | 0.7 m | - | U | < 15 min |
| Madeira | E | -16.666 | 32.750 | 51 | ~ 30 min | 0.9 m | 1.4 m | 2.5 m | - | U | ~ 30 min |
| | S | -16.926 | 32.619 | 51 | ~ 40 min | 0.3 m | 1.1 m | 2.1 m | - | U | ~ 30 min |
| Azores | | -27.150 | 38.800 | 53 | ~ 1 h 45 min | 0.5 m | 2.4 m | 4.2 m | - | U | ~ 15 min |
| Barbados | | -59.566 | 13.033 | 50 | ~ 7 h | 0.1 m | 0.6 m | 1.1 m | 0.45 – 0.6 m | U | ~ 30 min |

## 6. Discussion

We investigated possible sources of the earthquake and tsunami on the 31st March 1761 earthquake in the Atlantic.
Firstly, we excluded the locations similar to the instrumental events of the 20th century: 25.11.1941 (Baptista et al., 2016) and 26.05.1975 (Kaabouben et al., 2009) because of incompatibility of tsunami travel times (Fig.2).

Secondly, we placed a source about 85 km to the east of the location proposed by Baptista et al. (2006) (Fig. 2).

After setting the source position, we investigated focal mechanisms for the parent earthquake. We selected two focal mechanisms for testing: A and B. Solution A-MS corresponds to focal mechanism A with a multi-segment fault plane as
described in section 4.2 (table 2).

Our tests produce a set of TTTs compatible with the observations; Maximum differences between observed and predicted travel times are 15 minutes in the near-field and 30 minutes in the far-field. These differences are acceptable considering that the exact location of the observation points is unknown. Travel time results are valid for A, B and A-MS as the locations are the same. Tables 3 and 4 show that the predicted travel times are compatible with a source located in the area west of the Coral
Patch.

Any source located in the Northeast Atlantic south of the Scilly islands produces a shorter tsunami travel time to Scilly island than Mount's Bay. This fact shows that the 6 hours TTT reported in Kinsale contradicts the 4 hours TTT reported for Dungarvan (Fig. 1). On the other hand, the tsunami travel times predicted by our numerical simulation are consistent with their position related to the source area. The proposed source A produces wave heights applying the Green's Law to the values

recorded at the VTGs which are compatible with the observations in Lisbon, Kinsale, Scilly and Barbados (Fig. 5 and table 3). The results of the synthetic wave records of Dungarvan, Madeira and the Azores are compatible with the observations. In Mount's Bay, the wave height computed using the Green's Law of the VTG value is smaller than the one reported. However, analysis of figure 4 shows that the computed maximum wave heights greater than 1.6 m for Mount's Bay. This value agrees with the observation.

The proposed source B produces wave heights compatible with the observation in Lisbon, Scilly and Mount's Bay. We apply the Green's Law (Eq. 1) using the wave heights recorded at the VTG in Kinsale and Barbados and obtain larger wave heights than reported (table 4). Also, the computed maximum wave heights in figure 6 are higher than 1.4 m, 2.2 m and 0.7 m for Kinsale, Scilly and Barbados respectively. These values are higher than the one observed. At the Azores, the wave height reaches 4.2 m (table 4); however, the descriptions do not report an inundation. Also, at the coast of Morocco, source B predicts

wave heights close to 14 m. To our knowledge, the historical documents do not report any abnormal movement of the sea in Morocco.

The observations do not account for inundation in Lisbon. To investigate this fact, we estimated the tide condition in Lisbon for the 31st March 1761day. To do this, we used a Moon Phase table (USNO, 2017) and concluded that the tide was 2.6 m above hydrographic zero (HZ) (in dropping tide conditions) at 1 p.m. on the 31st of March 1761 (table 5).

The maximum of the synthetic wave record for source A-MSis 1.8 m about 2 hours and 15 minutes when the tide has dropped underneath 2.3 m above HZ. Adding 1.8 m to 2.3 m, we obtain 4.1m; this value is less than the tide amplitude in spring tide condition. Considering that Lisbon downtown was rebuilt 3 m above sea level after the 1755 event (Baptista et al., 2011) the predicted wave heights are compatible with no flooding.

The proposed source B generates for the first wave height 0.9 m but a maximum wave height of 2.2 m. The maximum wave

height occurs at 15:00 o'clock and the estimated tide is approximately 2.1 m above HZ. Adding 2.2 to 2.1 we reach spring tide condition of 4.3 m.

Given the considerations above the tide, analysis favours solution A.

**Table 5. Tide levels at the time of the earthquake and tsunami arrival.**

|  | Time | Tide condition | Estimated height relative to Hydrographic Zero |
| --- | --- | --- | --- |
| Earthquake | Noon | Full tide | 2.9 m |
| Tsunami arrival time | 13:15 | Dropping tide | 2.6 m |
| Max. wave height Hyp. AMS | 14:15 | Dropping tide | 2.3 m |

| | | | |
|---|---|---|---|
| Max. wave height Hyp. B | 15:00 | Dropping tide | 2.1 m |

The tidal range in Barbados is about 1 m. This small range might favour the observability of small first waves at tsunami arrival. For source A, the first wave in Barbados is about 0.1 m which raises the question if people might have noticed the advance of the sea. Close to 9 o'clock, 2 hours after tsunami arrival, the positive peak in the VTG is higher than 0.2 m which

results in 0.4 m when estimating the wave height applying the Green's Law for 5 m depth close to the shore. The coeval sources report similar wave height values.

Also, for source B, the wave height is smaller than 0.1 m at the VTG at the time of tsunami arrival. About 45 minutes later the waves are large than 0.2 m. The maximum peak occurs ca. 2 hours after tsunami arrival at 9 o'clock. Because of the small tide amplitude in Barbados does not contribute to select among the two candidate sources.

The summary (Annual register, 1761) states that the waves seemed to abate but at 10 o'clock started again with higher intensity and lasted until the next morning - this observation of greater amplitudes some hours after tsunami arrival fits for both sources. However, the timings of increasing wave heights do not match.

In Cadiz, both sources produce the observed withdrawal. Both sources A and B predict a drawdown of 0.6 m and 0.4 m respectively. High tide in Cadiz is about 1 hour earlier than in Lisbon. Once the tide was in dropping conditions at the time of

the tsunami arrival a larger drawdown is more likely to be observed.

Considering the points discussed above, we conclude our preferred solution is A-MS. Following facts justify our choice:

• The candidate source in hypothesis A-MS is compatible with the geodynamic setting predicted by the NUVEL 1A model (DeMets et al., 1999). NE/SW compressive structures with comparable fault plane parameters have been identified close to the Coral Patch seamount (Fig. 1 and 3). The proposed structure is possibly propagating and reactivating the NE-SW striking

oceanic rifting fabric towards the epicentre suggested by Baptista et al. (2006). Nevertheless, we must stress that Martínez-Loriente et al. (2013) do not suggest an extension of the seismogenic structure at the CPF although no detailed multi-channel seismic survey has been carried out to the west of the CPF in the proposed source area.

• The wave heights produced by the numerical models are in better agreement with proposed source A-MS.

• Wave heights greater than 14 m produced by hypothesis B would result in a catastrophic scenario which is rather unlikely

and nor observed neither or reported. Also, 4.2 m wave height produced by hypothesis B in the Azores would have caused inundation, which has not been reported.

• Although both solutions follow our considerations for Lisbon, the wave heights generated by source A-MS seem to be more comparable to the observed fluctuation of 2.4 m than the wave heights produced by source B.

• The larger drawdown in Cadiz favours solution A-MS.

The reassessment of the reports of Barbados and Cadiz support the choice selected here. While the tsunami travel time for Barbados supports the source location, the fact that there were no inundation reports in Cadiz supports the magnitude and rupture mechanism proposed here.

**7. Conclusion**

• The source proposed here for the 1761 event is compatible with the tsunami observation dataset, the macro-seismic intensity data (Baptista et al., 2006) and with the geodynamic context of the area predicted by the kinematic plate model NUVEL 1-A.

• The source proposed here is located in the SWIM, an area of widespread compressive structures (Hayward et al., 1999), corresponding to a fault that extends from the western segment of the CPF towards the epicentre proposed by Baptista et al. (2006).

The investigation of each historical event in the area contributes to a better understanding of the structure of this diffuse plate boundary and ultimately leads to a better evaluation of the seismic and tsunami hazard. This study together with the study by Baptista et al. (2006) underlines the need to include the 1761 event in all seismic and tsunami hazard assessments in the Northeast Atlantic basin.

**Acknowledgements.** This work is funded by FCT (Instituto Dom Luiz; FCT PhD grant ref. **PD/BD/135070/2017**). The authors wish to thank the editor Ira Didenkulova and the reviewers Uri S. ten Brink, Ceren Özer Sözdinler,Sara Martínez-Loriente and one anonymous referee for their constructive comments and suggestions that greatly helped to improve this manuscript. Finally, the authors wish to thank Pedro Terrinha for his valuable advice and interesting discussions on the geological context of the area and Paul-Louis Blanc for the translation of the report in Journal Historique (1773).

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
