# Peer review of "Reanalysis of the 1761 transatlantic tsunami"

_Natural Hazards and Earth System Sciences, 2018_

## Referee Comment (RC1) · U. S. ten Brink (Referee) · 29 May 2018

Review of reanalysis of the 1761 transatlantic tsunami, by Wronna et al. for NHESS

Baptista et al. (2006) carried out a detailed analysis of the 1761 earthquake and tsunami using historical records of both earthquake shaking intensity, T phases, and wave heights and travel time reports of tsunamis. The present manuscript focuses only on the tsunami evidence, using most of the evidence reported in Baptista et al. It calculated marigrams for two sources at Baptista's preferred location with two sets of fault parameters (strikes of 76° and 254.5°, dips of 40° and 70°, and rakes of 135° and 45°, respectively). It then argues qualitatively that the travel time matches the reported tsunami travel times and that the wave amplitude fits better one of the two fault parameter sets. Unfortunately, this single test to distinguish between two sets of fault parameters is not sufficient for publication in a journal and may be better suited for a

thesis chapter, because it does not advance our knowledge beyond Baptista's paper. In addition, the manuscript is poorly written and requires lots of clarifications.

The paper should clearly state the motivation behind this work, its novelty relative to Baptista et al. (2006). There is no need to describe in detail tsunami observations, which were already outlined by Baptista et al., or to describe each result. Instead, the paper should explain the methodology better, justify the reasons for the choices of the modeling parameters, and highlight and discuss significant results, that will advance our knowledge.

Following are detailed comments:

1. Modeled sources: Why were the specific strikes, dips, and rake for the 5 sources in Table 2 chosen, and why not other fault parameters? Why are only 2 of the 5 sources listed in Table 2 discussed and not the others? 2. Can you provide a more quantitative/statistical measure why you prefer one of the sources over the other (or over the other hypotheses which were not presented?) 3. Hyp. A-MS: How are the 4 segments of the first source arranged? Adjacent to each other or spaced or oriented at different strikes? What is the slip on each segment? 4. Did you consider modeling marigrams in locations which did not report a tsunami (e.g., the U.S. East Coast, other Caribbean sites) to test whether the rupture parameters produce insignificant marigrams there? 5. Were there any observations from Morocco? 6. Wave height: Please define wave height, maximum peak, etc. Do these terms only represent the positive part above a nominal Mean Sea level? Why don't the numbers listed in the text often match the marigrams in Figures 5 and 7 (e.g., section , 5.1-Cadiz 1.8 m in text, 2.3 in marigram; section 5.2 – Kinsale >1.5 in text, <0.8 m in marigram; Terceira >5.5 m in text, 2.4 m in marigram)? The max. wave height in Table 4 does not match the marigrams for Scilly and Mount's Bay, Kinsale, Azores, and Barbados. Are some of the values read from the maps and not from the marigrams? 7. Observations relative to the tidal cycle: The only location where the total tidal range and time relative to the tidal cycle were considered was Lisbon. What about the other locations? How did the second and third

waves arriving at different times in the tidal cycle, match the observations? 8. The Barbados marigrams show a much higher wave height 1.5-2 hours after the first wave arrival, or 9 hours after the event. Which wave arrival would have been noticed by eye witnesses? 9. At what water depths were the marigrams calculated? Did they take into account harbor reverberations, which affect the observed wave periodicity? How did the nested grids work if the original grid from which the bathymetry was derived, was much coarser?

Text, figures, other suggestions 1. Table 1 showing the observations is almost unreadable. I had difficulty matching locations with the other columns. Also, the locations need geographical coordinates. 2. Figures 2, 3 and the inset of Figure 1 can be combined to one figure. In this figure, please mark the locations of the Ampere and Coral Patch Seamounts and Horseshoe Abyssal Plain and list in the figure caption all the abbreviations that appear on the figure. 3. There are newer determinations of the relative plate motion along the boundary (Nocquet and Calais, 2004; Fernandes et al., 2007). Please mark the convergence vector from plate kinematics on your tested fault strikes. 4. Give a brief explanation of Mansinha and Smiley equations. Tsunami models typically use the Okada equations. 5. Were the time zone in Portugal, Portuguese Islands, the U.K. and Barbados similar to those today? Did every location measure their time independently relative to the sun's angle in the sky (i.e., latitudinal)? How well could minutes be measured in 1761? 6. Section 3 -There is no need to provide a verbal description of all the observations. They appear in Table 1 and Baptista et al. (2006). 7. There is no need to describe all the results of the synthetic tests in the text (p. 8-13). We can read them from the graphs. Describe only the most important points that you want the reader to pay attention to. 8. The reader is lost in the current discussion, which mixes lots of facts listed in a location by location list. 9. The final conclusion points are poorly written and confusing: What are "the area where there are the largest compressive structures"? Why is the timing of Barbados an important conclusion when the paper does not search for the best source location? Where was the 14 m wave height calculated? It was not mentioned earlier. 10. Blaser et al. not

Blazer. Withdrawal (noun) not withdraw.

---

## Referee Comment (RC2) · C. Ozer Sozdinler (Referee) · 9 Jul 2018

This manuscript mainly describes the studies performed for the verification of location of 1761 earthquake's epicenter using the observations at some coastal locations in SW Portugal, Caribbean Sea and Azores Islands recorded in historical documents. They apply a method for the relocation of the fault and propose various hypotheses for fault models having different fault parameters. After comparing with the tsunami numerical modeling results with the observed values, they conclude that the proposed source of 1761 earthquake is reliable and it should be included in the tsunami hazard-related studies in NE Atlantic Ocean. Of course, it is not possible to have 100% compatibility between the modeling results and observations because of many reasons such as unreliable historical records, not using so fine-gridded data for the smallest domain, using Green's Law for some VTGs, etc.

[Figure]

According to the proposed phenomenon and studies performed, this manuscript seems acceptable with minor revisions.

The most significant revision is needed for the idea of drawing of Euler Circle and defining the fault parameters accordingly. Since it is the basic of all this study, this part should be described more clearly and comprehensively.

The second revision should be for further description of backward ray tracing contours. This part is not clear to me; further details are needed for the meaning of these contours.

The other important revision is necessary for the comparison of observed data with the calculated results. The summary of results for 2 selected hypotheses are given in Tables 3 and 4 but there is no information for the observed wave heights at these locations. Instead, these values are given in Table 1. Table 1 may stay as it is but Table 3 and 4 should also include the observed values in a column for better comparison.

Another revision is recommended for giving further details regarding Paleo DEM mentioned on Page 8 very shortly. Since the modeling results may be affected due to such data, it is necessary to make further explanation on how you prepared/used this data and also its difference from the current DEM data.

Other minor corrections (typo, disorder meaning, etc) are given in lines with page numbers below:

- Page 1 Line 14: the phrase "...from Cadiz not used before" is not clear. - Page 2 Line 12: what does " we revisit the source..." mean? - Page 2 Line 15: Better to say "compared with" instead of "checked against" - Page 6 Line 2: It should be "... did not use it in the simulations...". "in" is missing. - Page 6 Line 15: Please rephrase the sentence "In a summary by Borlase (1762) summary describes..." - Page 6 Line 19: better to write 6 pm in numbers - Page 6 Line 31: In which region are these river estuaries located? - Page 8 Line 8: "...observation points.." instead of "...observations points..."

[Figure]

- Page 8 Line 25: The message of this sentence is not clear. Further explanation and clarification are needed. - Page 9: The first paragraph is a bit irrelevant with the previous and following ones. Better to link this paragraph with the previous one. - Page 9 Line 11: Better to say ". . . Figures from 4 to 7 present. . ." without using comma - Page 9 Line 14: Please rephrase the sentence "The geographical coordinates and depths their coordinates and depth are given. . ." - Page 10 Line 2: Please don't use comma after 5 - Page 10 Line 9: ". . .heights reach up to 1.7m" - Page 11 Line 11: better to use " leading elevation wave" instead of " an upward movement" -

The corrections for Figures and Tables are listed below: - Figure 1: Who suggested the other 2 epicenters of 1761 eq, except Baptista etal (2006)? Are they the ones also shown in Figure 2? If yes, then it is better to write them in Figure 1. Also, what are the lines with small black triangles represent in the zoomed-in map? It was not indicated in the legend. - Figure 2: In the caption, better to write "backward ray tracing" instead of "back ray.." - The plots in (b) and (c) of Figures 5 and 7 are not visible! They can be plotted with longer x-axis or separately one under the other with shorter y-axis. - Tables 3 and 4 should include historical tsunami observations at these locations in a different column - Page 13 Line 4: better to use "withdraw" instead of "downward movement"; "occurs" instead of "arrives" - Page 13 Line 5: better to use "water surface elevation" instead of "upward movement" - Page 13 Line 6: "wave ascending" instead of "upward movement" - Page 13 Line 7: ". . .waveform shows around 15 minutes wave period." - Page 13 Line 8: something missing here ". . .wave arrives at the _____ after 4 hours.." - Page 13 Line 8: ". . .15-minute period and 0.6m wave height" is better - Page 15 Line 9: Better to use word "delays or time difference" instead of "error" - Page 15 Line 19: Please rephrase the sentence starting with "Our source. . ." - Page 16 Line 20: Please rephrase this sentence; it is not clear.

The following references are not listed in the reference list: - Gutenberg and Richter (1949 - Moreira (1984) - DeMets etal (1990)

I believe the manuscript would be in a more comprehensive and well-understood

status if those revisions would be applied accordingly.

Please also note the supplement to this comment:
https://www.nat-hazards-earth-syst-sci-discuss.net/nhess-2018-30/nhess-2018-30-RC2-supplement.pdf

––––––––––––––––––––––––––––––––––––

---

## Author Comment (AC1) · 8 Aug 2018

Answers to comments U. S. ten Brink

Comment #1: The paper should clearly state the motivation behind this work, its novelty relative to Baptista et al. (2006). There is no need to describe in detail tsunami observations, which were already outlined by Baptista et al., or to describe each result. Instead, the paper should explain the methodology better, justify the reasons for the choices of the modeling parameters, and highlight and discuss significant results, that will advance our knowledge.

Answer #1: Up to Baptista et al. (2006) the 1761.03.31 earthquake was supposed to be located close to Galicia margin, based mainly on the interpretation of seaquake information. Baptista et al. (2006) made a revision of tsunami arrival times and macroseismic data to propose a preferred location at the SW Iberian margin. This conclusion was

reinforced by a manuscript describing the tsunami effects at Cadiz as described in this manuscript. This new paper aims at understanding the mechanism of this earthquake, in the framework of the tectonic setting of the South West Iberian Margin and the identification of the location of the plate boundary between Eurasia and Nubia close to the Strait of Gibraltar. The identification of a distinct fault plane is not straightforward outside active subduction zones. The focal mechanism of the tsunamigenic earthquakes located in the area that includes the South West Iberian Margin and the Gulf of Cadiz is one of these examples. Previous studies on the 1761 event do not investigate a possible earthquake mechanism compatible with the generation of a transatlantic tsunami. Also, the investigation of each of these events will contribute to better understand this diffuse plate boundary. We changed a part in the abstract and in the introduction to underline the objective of the study. Please see the changes in the abstract and the introduction in section 1.

The manuscript in the abstract now reads:

"Abstract. The segment of the Africa-Eurasia plate boundary between the Gloria fault and the Strait of Gibraltar has been the set of significant tsunamigenic earthquakes. However, their precise location and rupture mechanism remains poorly understood. The investigation of each event contributes to a better understanding of the structure of this diffuse plate boundary and ultimately leads to a better evaluation of the seismic and tsunami hazard. The 31st March 1761 event is one of the few known transatlantic tsunamis. Macroseismic data and tsunami travel times were used in previous studies to assess its source area. However, no one discussed the geological source of this event. In this study, we present a reappraisal of tsunami data to show that the observations dataset is compatible with a geological source close to Coral Patch and Ampere seamounts. We constrain the rupture mechanism with plate kinematics and the tectonic setting of the area. This study favors the hypothesis that the 1761 event occurred southwest of the likely location of the 1st November 1755."

The manuscript in section 1, the Introduction, now reads:

"In this study, we investigate the geological source of the 1761 transatlantic tsunami. To do this, we start with a reappraisal of previous research, we analyze the tectonic setting of the area and draw a source compatible with plate kinematics. From this source, we compute the initial sea surface displacement. To propagate the tsunami, we build a bathymetric dataset based on GEBCO (2014) data to compute wave heights offshore the observations points presented in table 1. We also compute inundation using high-resolution digital elevations models in Lisbon and Cadiz to check the results with the observations. Finally, we use Cadiz and Lisbon observations in 1755 and 1761 to compare the size of the events."

Comment #2: Modeled sources: Why were the specific strikes, dips, and rake for the 5 sources in Table 2 chosen, and why not other fault parameters?

Answer #2: In section 4.2 "Testing the hypothesis", we explain the choice of the location and describe how we approximated the size of the seismic structure using scaling laws. We find three candidate sources compatible with the scaling laws of Wells and Coppersmith (1994), Manighetti et al. (2007) and Blaser et al. (2010). We also state how we find the rake of the proposed faults, according to the difference between the strike and the velocity vector. However, we agree with the referee that the choice of all the parameters is not clearly explained and so the manuscript needs to be changed accordingly.

The manuscript now reads in section 4.2:

"4.2 Testing the hypothesis In the 20th century, two strong magnitude earthquakes occurred in the Gloria Fault (GF) area. Given this, we tested the compatibility of the tsunami observations in 1761 with the tsunamis produced by the earthquakes of the 25th November 1941 (Lynnes and Ruff, 1985; Baptista et al., 2016) and 26th May 1975 (Kaabouben et al., 2009). We use the fault plane parameters and rupture mechanism presented in Baptista et al. (2016) and Kaabouben et al. (2008) for the 1941 and 1975 events respectively. The fault dimensions and slip were made compatible with

an 8.5 magnitude event using the scaling laws proposed by Wells and Coppersmith (1994), Manighetti et al. (2007) and Blaser et al. (2010). These two events produce less than one-meter wave height in the North East Atlantic and were barely observed in the Caribbean Islands (Baptista et al., 2016; 2017). Moreover, the epicenters of the 25th November 1941 and 26th May 1975 are located outside the area determined by Baptista et al. (2006). As expected, the TTTs do not agree with those reported in 1761; therefore, we excluded the GF as a candidate source for the 1761 event and do not consider their results for discussion. The candidate fault area is centered at 12.00 W, 35.00 N to the west of the large NE/SW striking compressive structures (Martinez-Loriente et al., 2013) and 85 km northeast of the epicenter suggested by Baptista et al. (2006) (Fig. 3). We considered the fact that the historical accounts indicate an earthquake and tsunami less violent than 1755. To account for this, we used the fault dimensions presented in table 2 corresponding to a magnitude 8.4-8.5 earthquake (Baptista et al., 2006); consequently, the wave heights in Lisbon and Cadiz are smaller than those observed in the 1755 tsunami (Baptista et al., 1998). The fault dimensions presented in table 2 are compatible with the scaling laws of Wells and Coppersmith (1994), Manighetti et al. (2007) and Blaser et al. (2010). Hypotheses A and A-MS: Here we use a strike angle compatible with the study by Martinez-Loriente et al., (2013) that follows the morphology of the Coral Patch seamount (Fig. 1). The velocity vector predicted by NUVEL 1A (Fig. 3) together with the short periods (4-12 minutes) reported in 1761 (table 1) are in line with the mean dip angle of 40 degrees suggested by Martinez-Loriente et al. (2013) (table 2). We approximate the rake angle according to the difference between the convergence arrow given by the circle around the Euler Pole and the fault plane (Fig. 3). The wave period in Lisbon produced by this candidate source is close to 30 minutes. This value is not compatible with the observations (Table 1). To solve this problem, we implemented a multi-segment fault here called A-MS. This multi-segment solution consists of 4 segments each 50 km. The four segments are placed adjacent to each other, and the rupture mechanism is equal for each segment as in hypothesis A with a mean slip of 11m (Table 2). The

slip of each segment is presented in table 2. The synthetic waveforms are presented in figure 5 and discussed in sections 5 and 6. Hypothesis B: Finally, we test an alternative hypothesis B with a larger strike-slip component compared to hypothesis A. This also results in larger fault length and a steeper dip angle. Here, we consider a rupture along a fault plane rotated about 180° when compared to hypothesis A. To do this, we selected compatible strike and rake angles that results in a sinistral inverse lateral rupture (table 2). The synthetic waveforms are presented in figure 7 and discussed in sections 5 and 6."

Please find the changes in table 2 in the supplement.

Comment #3: Why are only 2 of the 5 sources listed in Table 2 discussed and not the others? Define wave height: The wave height is referenced to the still water level.

Answer #3: We agree with the referee that it is not evident in the discussion why we discard the other candidate sources. The manuscript needs to be changed accordingly. Please see the altered manuscript in section 4.2 in answer #2. We present the definition of wave height (referred to the still water level) in the paper in section 5, page 10, line 12. However, this definition must be introduced before table 1.

The text in section 3, paragraph 8 now reads:

"Table 1 presents a summary of all historical data relevant to the tsunami simulation. Figure 1 shows the locations of the tsunami observations. Wave heights always refer to the maximum positive amplitude above the still water level."

Comment #4: Can you provide a more quantitative/statistical measure why you prefer one of the sources over the other (or over the other hypotheses which were not presented?)

Answer #4: We consider two sources; sources A and B. Source A is compatible with the geodynamic setting of the area. Further, most of our results match with the observations. We discuss in section 6 why we favour Hypothesis A-MS. Also, the study

by Martinez-Loriente et al., (2013) suggest fault parameters like in Hypothesis A-MS. However, a statistical measure is somehow redundant when comparing our cases to the historical observations which inherently may have some error.

Comment #5: Hyp. A-MS: How are the 4 segments of the first source arranged? Adjacent to each other or spaced or oriented at different strikes? What is the slip on each segment?

Answer #5: We will follow the referees' suggestion and add the necessary information on the parameters and dimensions in table 2. Also, we will improve the description of how the segments of the faults are located. We change the text accordingly. Please see the altered manuscript of section 4.2 in answer #2.

Comment #6: Did you consider modeling marigrams in locations which did not report a tsunami (e.g., the U.S. East Coast, other Caribbean sites) to test whether the rupture parameters produce insignificant marigrams there?

Answer #6: We focus only on the sites where there are observations. We compute a marigram close to Anegada, to investigate the possibility of inundation which could be related with sediment layers of marine origin found by Atwater et al., (2012). Computed values are similar to those predicted for Barbados.

Comment #7: Were there any observations from Morocco?

Answer #7: To our knowledge, there were no observations in Morocco. We added the information in the manuscript in the discussion, section 6.

The corresponding paragraph now reads:

"Source B produces wave heights compatible with the observation in Lisbon, Scilly and Mount's Bay. We apply the Green's Law using the wave heights recorded at the VTG in Kinsale and Barbados and obtain larger wave heights than reported (table 4). Also, the modelled maximum wave heights in figure 6 are greater than 1.4 m, 2.2 m and 0.7 m for Kinsale, Scilly and Barbados respectively. These values are greater than

the one observed. At the Azores, the wave height reaches 4.2 m (table 4); however, the descriptions do not report an inundation. Also, at the coast of Morocco, source B predicts wave heights close to 14 m. To our knowledge, the historical documents do not report any abnormal movement of the sea in Morocco."

Comment #8: Wave height: Please define wave height, maximum peak, etc. Do these terms only represent the positive part above a nominal Mean Sea level?

Answer #8: Please see answer #3.

Comment #9: Why don't the numbers listed in the text often match the marigrams in Figures 5 and 7 (e.g., section, 5.1-Cadiz 1.8 m in text, 2.3 in marigram; section 5.2 – Kinsale >1.5 in text, <0.8 m in marigram; Terceira >5.5 m in text, 2.4 m in marigram)? The max. wave height in Table 4 does not match the marigrams for Scilly and Mount's Bay, Kinsale, Azores, and Barbados. Are some of the values read from the maps and not from the marigrams?

Answer #9: For Lisbon and Cadiz, where high-resolution bathymetric data was available we used this data to build a system of nested grids to compute the wave height close to shore. For all the other sites where no high-resolution data were available, we set the tide gauge in deeper water and extrapolate wave heights using the Greens Law to a fixed water depth of 5 m.

We agree with the referee and changed the manuscript accordingly. We clarify where we use nesting and where we compute nearshore wave heights according to the Greens Law. For consistency, we also changed in the tables 3-4 the wave height values in the columns "first" and "max." before extrapolation according to the Greens Law. These values are now coherent with the values in the marigrams. In tables 3-4 we added an extra column where we present the maximum wave height after application of the Greens Law. In section 4.1 we add a paragraph explaining the Greens Law.

We changed the text and tables accordingly. The text and tables now read in section

4.1 and section 5:

"4.1 The numerical model We use the code NSWING (Non-linear Shallow Water model wIth Nested Grids) for numerical tsunami modeling. The code solves linear and non-linear shallow water equations (SWEs) in a Cartesian or spherical reference frame using a system of nested grids and a moving boundary condition to track the shore-line motion based on COMCOT (Cornell Multi-grid Coupled Tsunami Model; Liu et al., 1995; 1998). The code was benchmarked with the analytical tests presented by Syn-olakis et al. (2008) and tested in Miranda et al. (2014) and Baptista et al. (2016), Wronna et al. (2015) and Omira et al. (2015). For Cadiz and Lisbon only, where high-resolution bathymetric data was available, we employ a set of coupled nested grids with a final resolution of 25 m to compute inundation. We compute a new bathymetric dataset using the nautical charts close to the coast or LIDAR data to build a Digital Terrain Model to compute inundation in Lisbon and Cadiz. Close to the tsunami source we interpolate of the source area bathymetry (GEBCO, 2014) to obtain a 1600 m grid cell size. We apply a refinement factor of 4 for the four nested grids. Consequently, the intermediate grids have a resolution of 100 m and 400 m respectively. In Cadiz, we use the soundings and coastline of historical nautical charts from the 18th century (Bellin, 1762 and Rocque, 1762) to compute a Paleo Digital Elevation Model (PDEM) (Wronna et al., 2017). To do this we geo-reference the old nautical charts and use the modern-day DEM (UG-ICN, 2009) to implement the information from the ancient charts. According to Wronna et al. (2017) we systematically remodel bathymetry and the coastline. To initiate the tsunami propagation model, we compute the co-seismic deformation according to the half-space elastic theory (Mansinha and Smylie, 1971) implemented in Mirone suite (Luis, 2007). Assuming that water is an incompressible fluid we translate the sea bottom deformation to the initial sea surface deformation and set the velocity field to zero for the time instant t = 0 s. We ran the model for 10-hour propagation time to ensure that the tsunami reaches all observation points. We compute the offshore wave heights for points located close to the observation points (Fig. 1) using Virtual Tide Gauges (VTG). We include the coordinates and depths of the

VTG in the tables 3 and 4 in section 5. For transatlantic propagation, we consider the Coriolis effect in the tsunami simulation. All tsunami simulations were checked against historical data. For the locations in Ireland, the United Kingdom, the Azores, Madeira and Barbados we use the approximation according to the Greens Law (Green, 1838). The Greens Law is based on the linear shallow water wave equations and allows to quickly approximate the amplification of wave heights at a shallower depth close to the shore when considering a plane beach. The wave height increases to the fourth root of the ratio between the depth at the shore and the water depth at the VTG. We extrapolate the maximum wave height values between the depths of the VTG (table 3 and 4) to points located at 5 m depth. $h\_s=∜(d\_s/d\_d)*h\_d$ Eq. (1) Where h_s and h_d are the wave heights at the shore and the VTG respectively, and d_s and d_d are the depths at the shore and the VTG respectively. For d_s we use a constant value of 5 m. the results of the approximation according to the Greens Law are presented in table 3 and 4."

Please find the changes in the tables 3 and 4 in section 5.1 and 5.2 in the supplement

Comment #10: Observations relative to the tidal cycle: The only location where the total tidal range and time relative to the tidal cycle were considered was Lisbon. What about the other locations? How did the second and third waves arriving at different times in the tidal cycle, match the observations?

Answer #10: The tidal range in Barbados is small, reaching approximately 3 feet → tsunami amplitude 2 feet. Observations in northern Europe confirm that the tide should fall still for another hour when tsunami arrived. We agree with the referee changed the discussion accordingly.

The manuscript in the discussion, section 6 now reads:

[revised manuscript text omitted]

Please find the changes in table 5 in the supplement.

Comment #11: The Barbados marigrams show a much higher wave height 1.5-2 hours after the first wave arrival, or 9 hours after the event. Which wave arrival would have been noticed by eye witnesses? Repeat eyewitness accounts.

Answer #11: We agree with the referee and add the information in the discussion, section 6. Please find the changed discussion, section 6 in answer #3.

Comment #12: At what water depths were the marigrams calculated? Did they take into account harbor reverberations, which affect the observed wave periodicity? How did the nested grids work if the original grid from which the bathymetry was derived, was much coarser?

Answer #12: The depths of the virtual marigrams are given in tables 3 and 4. They do not take into account harbor reverberation. Most of the tide gauges are placed outside the harbors at a depth of 50m at the continental platform. For Lisbon, we positioned the observation point to compute the synthetic marigram tide gauge in the Tagus estuary close to the city center. For Cadiz where high resolute bathymetric data was available, we also use nested grids and place the virtual tide Gauge close to the shore. Please find the coordinates and the depth of the VTG in table 3 and 4. We use the nested grids only for Lisbon and Cadiz where we have better bathymetry data digitized from nautical maps. In all other places, we use a rectangular grid with data obtained from GEBCO. In these points, we do not use nested grids in the simulation. We now better explain in section 4.1 where we use nested grids and where we apply the Greens Law.

Please find the changed section 4.1 in answer #9.

Text, figures, other suggestions

Comment #13: Table 1 showing the observations is almost unreadable. I had difficulty matching locations with the other columns. Also, the locations need geographical coordinates.

Answer #13: We agree with the referee and added shading to enhance the readability of the table. We also introduced geographical coordinates.

Please find the changes in table 1 in the supplement.

Comment #14: Figures 2, 3 and the inset of figure 1 can be combined to one figure. In

this figure, please mark the locations of the Ampere and Coral Patch Seamounts and Horseshoe Abyssal Plain and list in the figure caption all the abbreviations that appear on the figure.

Answer #14: The convergence arrow according to Nuvel 1A is plotted along the fault in figure 3. We consider the suggestion of the referee but find that two figures separately enhance their readability.

Comment #15: There are newer determinations of the relative plate motion along the boundary (Nocquet and Calais, 2004; Fernandes et al., 2007). Please mark the convergence vector from plate kinematics on your tested fault strikes.

Answer #15: There are a few geodetic and geophysical plate models that describe the relative motion between the Eurasian and the Africa/Nubia plates. Nevertheless, for this work they are not significantly different. We chose NUVEL1A because it is a good and robust description of plate kinematics in the area. Nonetheless, we introduce additional references as suggested and the convergence vector used in this work is plotted in the corresponding figure.

The manuscript now reads in section 2, paragraph 3:

"Kinematic plate models (Argus et al., 1989; DeMets et al. 1999; Nocquet and Calais 2004; Fernandes et al., 2007) show low convergence rates 3 - 5 mm per year between African plates and Eurasia. We used the global kinematic plate model Nuvel-1A. This model is a recalibrated version of the precursor model Nuvel-1 that implements rigid plates and data from plate boundaries such as spreading rates, transform fault azimuths, and earthquake slip vectors (DeMets et al., 1990). The NUVEL 1A model predicts a relatively conservative convergence rate of 3.8 mm per year in the area close to the source area determined by Baptista et al. (2006) for the 1761 tsunami (Fig. 2). Consequently, we consider a possible fault as an extension of the CPF closest to the area presented by Baptista et al. (2006). We draw the circle around the Euler pole at -20.61 w, 21.03 N according to the plate kinematic model Nuvel 1-A using Mirone suite

(Luis 2007). To do this we chose Africa as fixed plate and Eurasia as moving plate and draw the circle at the center of the fault in figure 3. We compute the convergence rate (3.8 mm per year) and plot the tangent velocity vector along the circle (Fig. 3). For this fault, we test different earthquake fault parameters and compute the co-seismic deformation using the Mansinha and Smiley equations (Mansinha and Smiley, 1971). We assume that the initial sea surface elevation mimics the sea bottom deformation and we use it to initiate the tsunami propagation model "

Comment #16: Give a brief explanation of Mansinha and Smiley equations. Tsunami models typically use the Okada equations.

Answer #16: The static deformation of the ocean bottom used to compute the initial sea surface displacement is deduced by the analytical formulae of Mansinha and Smiley 1971 or Okada 1985. For slip along a rectangular fault in a homogeneous half-space the static deformation (co-seismic displacement) can be obtained by the analytical formulae of Mansinha and Smiley or Okada equivalent expressions. The difference between these two formulations is that Okada's allows for non-double couple solutions.

Comment #17: Were the time zone in Portugal, Portuguese Islands, the U.K. and Barbados similar to those today? Did every location measure their time independently relative to the sun's angle in the sky (i.e., latitudinal)? How well could minutes be measured in 1761?

Answer #17: The problem of the solar times in the 18th century is addressed in Baptista et al., 1998. Here we followed the same procedure. The document by Torres Vilarruel (1756) quoted by Baptista et al. 1998 clearly states the difference in time between Lisboa and Madrid. Here we used the solar time differences computed by Observatório Nacional da Ajuda (Lisboa, Portugal) and presented in Baptista et al., (1998).

We add this information to the manuscript. Section 3, paragraph 3 now reads:

"We assume as in Baptista et al. (1998a, b) that all times are solar time and we re-evaluate the Tsunami Travel Time (TTT) for Barbados. For Barbados, documents report a tsunami arrival at a 4 pm local time. Baptista et al. (2006) concluded for the unreliability of this observation and did not use it for the simulations to locate the source. In this study, we use 3.5 hours solar time difference between Lisbon and Barbados. Using 4 pm local time as stated in Borlase (1762) for the arrival of the tsunami and the 3.5 h solar time difference between Lisbon and Barbados, we conclude a TTT of 7-7.5 h. We place a point source at Barbados and use backward ray tracing and find that the 7 h contour falls within the area presented by Baptista et al. (2006) close to their suggested location (Fig. 1 and 2) at 34.50 N 13.00 W. Mason (1761) wrote that the tide ebbed and flowed between eighteen inches and two feet."

Comment #18: Section 3 -There is no need to provide a verbal description of all the observations. They appear in Table 1 and Baptista et al. (2006).

Answer #18: We believe that some verbal description helps the reader to understand the historical observation better. However, we think that the description should be short and concise. We maintain the most important descriptions of the observations discussed in our results which we consider to be essential for the reader to understand. We changed the text accordingly.

The manuscript in section 3 now reads:

"3. Reassessment of historical data on the 1761 tsunami The studies by Baptista et al. (2006) and Baptista and Miranda (2009) present most of the tsunami information used herein. However, only the information on tsunami travel times was used by these authors to locate the source (Baptista et al., 2006). In this study, we reappraise the tsunami observations regarding tsunami travel time and wave heights, period and duration of the sea disturbance. For Cadiz, The Journal des Matiéres du Temps (Journal Historique, 1773), describes the occurrence of an earthquake in April 1773 and compares it with the 31st March 1761 event. The document states that in April 1773, following an earthquake felt in Cadiz, it was feared that it could have triggered a tsunami. The governor of the city ordered the closing of the town gates to prevent people fleeing to the causeway which was inundated in 1755. The report concludes that no tsunami was observed in 1773. However, the text of the report suggests a withdraw of the sea after the 31st March 1761 earthquake in the city. We assume as in Baptista et al. (1998a, b) that all times are solar time and we re-evaluate the Tsunami Travel Time (TTT) for Barbados. For Barbados, documents report a tsunami arrival at a 4 pm local time. Baptista et al. (2006) concluded for the unreliability of this observation and did not use it for the simulations to locate the source. In this study, we use 3.5 hours solar time difference between Lisbon and Barbados. Using 4 pm local time as stated in Borlase (1762) for the arrival of the tsunami and the 3.5 h solar time difference between Lisbon and Barbados, we conclude a TTT of 7-7.5 h. We place a point source at Barbados and use backward ray tracing and find that the 7 h contour falls within the area presented by Baptista et al. (2006) close to their suggested location (Fig. 1 and 2) at 34.50 N 13.00 W. Mason (1761) wrote that the tide ebbed and flowed between eighteen inches and two feet. For Lisbon, the reports state abnormal motion of the sea about 1 hour and 15 minutes after the earthquakes. Two sources (Unknown, 1761 and Molloy, 1761) describe a flowing and ebbing of 8 feet of about six minutes while Borlase (1762) reports only three to four feet. All three reports agree that the agitation lasted until the evening. The descriptions from northern Europe include Mount's Bay, Scilly Islands, Kinsale and Dungarvan (Table 1 and Figure 1). Borlase (1762) reports the tsunami observations at several points in Mount's Bay. The waves arrived around five o'clock in the afternoon at about one and a half hour before full ebb. According to the report, the water rose between four and six feet, and the sea advanced and recessed five times within an hour (Table 1). At Scilly Islands, the report states that the sea rose four feet and that the agitation lasted about 2 hours. In Kinsale, the Annual Register (1761) states that at 6 p.m. at low water, the tide rose quickly about two feet higher than it was and it ebbed again about four minutes later. The movement of the fluxes repeated several times but with decreasing intensity after the in and outflux. In Dungarvan, Borlase (1762) states

that the sea ebbed and flowed five times between 4 and 9 o'clock in the afternoon. Table 1 presents a summary of all historical data relevant to the tsunami simulation. Figure 1 shows the locations of the tsunami observations. Wave heights always refer to the maximum positive amplitude above the still water level."

Comment #19: There is no need to describe all the results of the synthetic tests in the text (p. 8-13). We can read them from the graphs. Describe only the most important points that you want the reader to pay attention to.

Answer #19: We agree with and changed the section 5 accordingly.

The manuscript in section 5 now reads:

[revised manuscript text omitted]

Please find the changes of the figures and tables of section 5 in the supplement.

Comment #20: The reader is lost in the current discussion, which mixes lots of facts listed in a location by location list.

Answer #20: We agree and changed the discussion accordingly. Please find the changed discussion and conclusion, section 6 in answer #10.

Comment #21: The final conclusion points are poorly written and confusing: What are "the area where there are the largest compressive structures"? Why is the timing of Barbados an important conclusion when the paper does not search for the best source location? Where was the 14 m wave height calculated? It was not mentioned earlier.

Answer #21: We agree and changed the manuscript accordingly. Please find the changed discussion and conclusion, section 6 in answer #10.

Comment #22: Where was the 14 m wave height calculated? It was not mentioned earlier.

Answer #22: 14 m wave height was computed at uninhabited area along the coast of Morocco for hypothesis B. Figure 6 shows this value. We changed the parts in the manuscript accordingly. Please see answer #7.

Comment #23: Blaser et al. Not Blazer. Withdrawal (noun) not withdraw.

Answer #23: We corrected the mistakes.

Please also note the supplement to this comment:
https://www.nat-hazards-earth-syst-sci-discuss.net/nhess-2018-30/nhess-2018-30-
AC1-supplement.pdf
2018-30, 2018.

---

## Author Comment (AC2) · 8 Aug 2018

Response to reviewer Ceren Sozdinler

Major comments:

Comment #1: The most significant revision is needed for the idea of drawing of Euler Circle and defining the fault parameters accordingly. Since it is the basic of all this study, this part should be described more clearly and comprehensively.

Answer #1: In plate kinematics the relative motion between two plates can be described by rotation around an Euler pole. We draw the circle around the Euler pole according to the kinematic global kinematic plate model Nuvel-1A. We chose as fixed plate Africa and as moving plate Eurasia. We draw the circle at the proposed location of the candidate fault at 12.00 W, 35.00 N around the Euler pole at -20.61 W, 21.03 N.

The global kinematic plate model computes a relative convergence rate of 3.8 mm per year.

We agree with the referee that more detailed explanation is necessary and adopt the corresponding sections in the manuscript accordingly.

The manuscript in section 2 now reads:

[revised manuscript text omitted]

Please find the changes in table 2 in the supplement.

Comment #2: The second revision should be for further description of backward ray tracing contours. This part is not clear to me; further details are needed for the meaning of these contours.

Answer #2: In section 3, paragraph 3 we describe how we obtain the contours in figure 2 and 3. The contours show a tsunami travel time of 7 and 7.5 hours respectively (Fig. 2 and 3). The prosed fault is located within these contours. Baptista et al. (2006) used macro seismic analysis and backward ray tracing and conclude a source area delimited by the orange contours in figure 2. Once the 7 hours contour falls within their proposed area and the observed tsunami travel time was 7 – 8 hours we propose a source in between the 7 h and 7.5 h contours.

We changed the corresponding paragraph in section 3.

The manuscript now reads:

"We assume as in Baptista et al. (1998a, b) that all times are solar time and we re-evaluate the Tsunami Travel Time (TTT) for Barbados. For Barbados, documents report a tsunami arrival at a 4 pm local time. Baptista et al. (2006) concluded for the unreliability of this observation and did not use it for the simulations to locate the source. In this study, we use 3.5 hours solar time difference between Lisbon and Barbados. Using 4 pm local time as stated in Borlase (1762) for the arrival of the tsunami and the 3.5 h solar time difference between Lisbon and Barbados, we conclude a TTT of 7-7.5 h. We place a point source at Barbados and use backward ray tracing and find that the 7 h contour falls within the area presented by Baptista et al. (2006) close to their suggested location (Fig. 1 and 2) at 34.50 N 13.00 W."

Comment #3: The other important revision is necessary for the comparison of observed data with the calculated results. The summary of results for 2 selected hypotheses are given in Tables 3 and 4 but there is no information for the observed wave heights at these locations. Instead, these values are given in Table 1. Table 1 may stay as it is but Table 3 and 4 should also include the observed values in a column for better comparison.

Answer #3: We agree with the referee and changed the tables accordingly.

Please find the changes in the tables 3 and 4 in the supplement.

Comment #4: Another revision is recommended for giving further details regarding Paleo DEM mentioned on Page 8 very shortly. Since the modeling results may be affected due to such data, it is necessary to make further explanation on how you prepared/used this data and also its difference from the current DEM data.

Answer #4: We agree with the referee and changed the manuscript accordingly.

The manuscript in section 4.1, paragraph 3 now reads:

"In Cadiz, we use the soundings and coastline of historical nautical charts from the 18th century (Bellin, 1762 and Rocque, 1762) to compute a Paleo Digital Elevation

Model (PDEM) (Wronna et al., 2017). To do this, we geo-referenced the old nautical charts and use the modern-day DEM (UG-ICN, 2009) to implement the information from the ancient charts. According to Wronna et al. (2017) we systematically remodel the bathymetry and the coastline. "

Minor comments:

Comment #5: - Page 1 Line 14: the phrase "...from Cadiz not used before" is not clear.

Answer #5: We changed the abstract.

The abstract now reads:

"The segment of the Africa-Eurasia plate boundary between the Gloria fault and the Strait of Gibraltar has been the set of significant tsunamigenic earthquakes. However, their precise location and rupture mechanism remains poorly understood. The investigation of each event contributes to a better understanding of the structure of this diffuse plate boundary and ultimately leads to a better evaluation of the seismic and tsunami hazard. The 31st March 1761 event is one of the few known transatlantic tsunamis. Macroseismic data and tsunami travel times were used in previous studies to assess its source area. However, no one discussed the geological source of this event. In this study, we present a reappraisal of tsunami data to show that the observations dataset is compatible with a geological source close to Coral Patch and Ampere seamounts. We constrain the rupture mechanism with plate kinematics and the tectonic setting of the area. This study favors the hypothesis that the 1761 event occurred southwest of the likely location of the 1st November 1755."

Comment #6: - Page 2 Line 12: what does "we revisit the source..." mean?

Answer #6: We revisit the source of the 1761 earthquake by summarizing the results of earlier studies and include new findings. However, we believe that the manuscript needs some alterations. We changed this paragraph of the manuscript.

The paragraph on page 2, line 10 now reads:

"In this study, we investigate the geological source of the 1761 transatlantic tsunami. To do this, we start with a reappraisal of previous research on TTT, we analyze the tectonic setting of the area and draw a source compatible with plate kinematics. From this source we compute the initial sea surface displacement. To propagate the tsunami, we build a bathymetric dataset based on GEBCO (2014) data to compute wave heights offshore the observations points presented in table 1. We also compute inundation using high resolution digital elevations models in Lisbon and Cadiz to compare the results with the observations. Finally, we use Cadiz and Lisbon observations in 1755 and 1761 to compare the size of the events. "

Comment #7: - Page 2 Line 15: Better to say "compared with" instead of "checked against"

Answer #7: Please see answer #6.

Comment #8: - Page 6 Line 2: It should be "...did not use it in the simulations...". "in" is missing.

Answer #8: We corrected the mistake.

The manuscript now reads:

"We use solar times at each observation point as in Baptista et al. (1998). Baptista et al. (2006) concluded for the unreliability of this observation and did not use it in the simulations to locate the source."

Comment #9: - Page 6 Line 15: Please rephrase the sentence "In a summary by Borlase (1762) summary describes..."

Answer #9: We rephrased the entire section 3.

Section 3, paragraph 5 now reads:

"For Lisbon, the reports state abnormal motion of the sea about 1 hour and 15 minutes after the earthquakes. Two sources (Unknown, 1761 and Molloy, 1761) describe a

flowing and ebbing of 8 feet of about six minutes while Borlase (1762) reports only three to four feet. All three reports agree that the agitation lasted until the evening. "

Comment #10: - Page 6 Line 19: better to write 6 pm in numbers

Answer #10: We agree and changed the manuscript accordingly.

The manuscript now reads in section 3, paragraph 7:

"At Scilly Islands, the report states that the sea rose four feet and that the agitation lasted about 2 hours. In Kinsale, the Annual Register (1761) states that at 6 p.m. at low water, . . .."

Comment #11: - Page 6 Line 31: In which region are these river estuaries located?

Answer #11: Borlase summary states uncommon motions in the river Sure in Carrick and Waterford and in the river Barrow in Ross. All sites are located in Ireland. However, we delete the sentence because it is not relevant in our study.

Comment #12: - Page 8 Line 8: "...observation points.." instead of "...observations points..."

Answer #12: We corrected the mistake.

Comment #13: - Page 8 Line 25: The message of this sentence is not clear. Further explanation and clarification are needed.

Answer #13: We agree and rephrased the sentence.

The sentence now reads:

"We considered the fact that the historical accounts indicate an earthquake and tsunami less violent than the 1755."

Comment #14: - Page 9: The first paragraph is a bit irrelevant with the previous and following ones. Better to link this paragraph with the previous one.

[Figure]

Answer #14: We believe it is worth to explain in the beginning of the results section how the section is structured and maintain the paragraph in the manuscript.

Comment #15: - Page 9 Line 11: Better to say "...Figures from 4 to 7 present ..." without using comma

Answer #15: We deleted the comma.

Comment #16: - Page 9 Line 14: Please rephrase the sentence "The geographical coordinates and depths their coordinates and depth are given..."

Answer #16: We rephrased the sentences.

The sentence now reads:

"The geographical coordinates and depths of the VTGs are given in tables 3 and 4."

Comment #17: - Page 10 Line 2: Please don't use comma after 5

Answer #17: We deleted the comma.

Comment #18: - Page 10 Line 9: "... heights reach up to 1.7m"

Answer #18: We rephrased section 5.1.

The sentence now reads:

"In Great Britain, at the Scilly Islands and Mount's Bay maximum wave heights vary between 1.7 and 1.9 m."

Comment #19: - Page 11 Line 11: better to use "leading elevation wave" instead of " an upward movement"

Answer #19: We adopted the manuscript according to the referees' suggestion.

The sentence now reads:

"All VTGs in northern Europe recorded the first wave as leading elevation wave (Fig. 5

(b)).”

Comments on Figures and Tables:

Comment #20: - Figure 1: Who suggested the other 2 epicenters of 1761 eq, except Baptista etal (2006)? Are they the ones also shown in Figure 2? If yes, then it is better to write them in Figure 1. Also, what are the lines with small black triangles represent in the zoomed-in map? It was not indicated in the legend.

Answer #20: We agree and include the information in figure 1. We also complete the information in the figure caption and legend.

Please see the changes in figure 1 in the supplement.

Comment #21: - Figure 2: In the caption, better to write “backward ray tracing” instead of “back ray..”

Answer #21: We changed the figure caption as suggested.

Comment #22: - The plots in (b) and (c) of Figures 5 and 7 are not visible! They can be plotted with longer x-axis or separately one under the other with shorter y-axis.

Answer #22: We agree and changed figures 5 and 7.

Please find the changes in figures 5 and 7 in the supplement.

Comment #23: - Tables 3 and 4 should include historical tsunami observations at these locations in a different column.

Answer #23: We agree and introduced and additional column. Please see answer #3.

Comment #24: - Page 13 Line 4: better to use “withdraw” instead of “downward movement”; “occurs” instead of “arrives”

Answer #24: We changed the manuscript according to the suggestion.

Comment #25: - Page 13 Line 5: better to use “water surface elevation” instead of

"upward movement"

Answer #25: We agree and adapted the manuscript according to the suggestion.

Comment #26: - Page 13 Line 6: "wave ascending" instead of "upward movement"

Answer #26: We rephrased the sentence.

The manuscript now reads:

"The first wave reaches 0.4 m, arriving close to 4 h after the earthquake."

Comment #27: - Page 13 Line 7: "... waveform shows around 15 minutes wave period."

Answer #27: We agree and adapted the manuscript according to the suggestion.

Comment #28: - Page 13 Line 8: something missing here "... wave arrives at the ____ after 4 hours.."

Answer #28: We rephrased the sentence.

The manuscript now reads:

"In Mount's Bay, the first wave of 0.4 m arrives after 4 hours and 30 minutes with a 15-minute period."

Comment #29: - Page 13 Line 8: "... 15-minute period and 0.6m wave height" is better

Answer #29: We agree and rephrased the paragraph in section 5.

The paragraph now reads:

"The maximum wave heights at the Scilly Islands is 0.5 m (Fig. 7 (b)). The first wave reaches 0.4 m, arriving close to 4 h after the earthquake. The synthetic tsunami waveform shows around 15-minute wave period. In Mount's Bay, the first wave of 0.4 m arrives after 4 hours and 30 minutes with a 15-minute wave period. Here, the maximum wave height, 0.7 m, comes more than 6 hours after the earthquake. In Kinsale, hypothesis B produces a maximum wave height of 0.6 m. The first wave of 0.2 m wave

height in the VTG arrives after 4 hours and 15 minutes of tsunami propagation; here, the period is shorter than 15 min (Fig. 7 (c))."

Comment #30: - Page 15 Line 9: Better to use word "delays or time difference" instead of "error"

Answer #30: We agree and changed the manuscript accordingly.

The manuscript now reads:

"Our tests produce a set of TTTs compatible with the observations with a 15-minute delay in the near-field and 30-minute delay in the far-field. These differences are acceptable considering that the location of the observation point is unknown."

Comment #31: - Page 15 Line 19: Please rephrase the sentence starting with "Our source ..."

Answer #31: We rephrased the corresponding paragraphs in section 6.

The manuscript now reads in section 6:

"Our tests produce a set of TTTs compatible with the observations with a 15-minute delay in the near-field and 30-minute delay in the far-field. These differences are acceptable considering that the location of the observation point is unknown. These results are valid for A, B and A-MS as the locations are similar. Tables 3 and 4 show that the predicted travel times are compatible with a source located in the area of the Coral Patch. Any source located in the Northeast Atlantic south of the Scilly islands produces a shorter tsunami travel time to Scilly island than Mount's Bay. The 6 hours TTT reported in Kinsale contradicts the 4 hours TTT reported for Dungarvan (Fig. 1). On the other hand, the tsunami travel times predicted by our numerical simulation are consistent with their relative geographical position."

Comment #32: - Page 16 Line 20: Please rephrase this sentence; it is not clear.

Answer #32: We rephrased section 6.

The corresponding paragraph now reads:

"The tidal range in Barbados is about 1 m. This small range might favor the observability of small first waves at tsunami arrival. For source A, the first wave in Barbados is about 0.1 m which raises the question if people might have noticed the advance of the sea. Close to 9 o'clock 2 hours after tsunami arrival, the peak at the VTG is higher than 0.2 m which results in 0.4 m when estimating the wave height applying the Green's Law for 5 m depth close to the shore. The coeval sources report similar wave height values."

Comment #33: -The following references are not listed in the reference list: - Gutenberg and Richter (1949) - Moreira (1984) - DeMets etal (1990)

Answer #33: We included the missing references.

Please also note the supplement to this comment:
https://www.nat-hazards-earth-syst-sci-discuss.net/nhess-2018-30/nhess-2018-30-AC2-supplement.pdf

---

## Author Response (AR2)

Prior consideration of the referee:

Tsunami modelling is not my field of research, so I think I should not evaluate that part of the work.

**Major comments**

**Comment 1:** Given that the innovation of this work with respect Baptista et al. (2006) is that here they propose a geological source for the 1761 earthquake (EQ.), the reader expects to find a strong geological background in this paper supporting their proposal ... and this is not the case. Some important references of the study area should be added to the references list, especially the works focused on the study of tectonic structures mentioned throughout the work (and others missing!). The main problem that I detect in this work is that the proposed geological source for the 1761 EQ does not exist.

**Answer 1:** We appreciate the referee comments focusing on the geologic background and agree with Sara Martínez-Loriente's suggestions and added the references accordingly. Also, we completed the present-day geological knowledge of the area and added a geological justification for the choice of the proposed hypothetical fault. To underline our choices, we ameliorated figure 3.

The relationship between seismic and geological sources is quite difficult in the SWIM area, even in the case of instrumental events (for example 28th Feb. 1969). This is a possible consequence of the distributed deformation associated with the slow converging plate boundary. The best contribution to shed light on the characterisation of tsunami (and seismic) hazard in SW Iberian is to use tsunami observations to constrain the location of an active structure. Having said that, the goal of this study is the identification of tectonic features that might justify the tsunami observations of the 31st March 1761; additionally, the source proposed here must be compatible with the area presented by Baptista et al. (2006). Similarly, to the study performed by Baptista et al. (1998) for the 1755 event, here we identify the most likely location and mechanism compatible with the observations. The identification of the scope of our study. The modelling results presented in the manuscript inherit limitations because of the paucity of data and interpretation that it is not always straightforward. However, we were able to present a solution that succeeds in reproducing most of the tsunami observations.

We propose a source location and rupture mechanism compatible with: the results of Baptista et al. (2006) and the set of tsunami travel times and wave heights. We place the candidate source North of the Coral Patch seamount where compressive structures were identified by Hayward et al. (1999) using shallow seismic reflection profiles and side scan sonar data. These structures are also shown in Zitellini et al. (2009).

**The text in section 2 now reads:**

[revised manuscript text omitted]

New figure 3 is presented below:

---

## Author Response (AR3)

**Answers to the anonymous referee:**

**Main concerns and suggested changes:**

[As I am not very familiar with the geology of this area, my comments are generally focussed on the tsunami modelling aspects]

My primary concerns are to do with limitations in the source identification methodology, which I would like to see more clearly acknowledged.

**Comment 1:** The authors set out to distinguish between source models which vary according to the extent of strike-slip motion included. Their more-compressive model A and a more-strike-slip model B. As the original model A produced wave periods inconsistent with historical observations they implemented a four-segment model 'A-MS' with varying slip of 7/15/15/8 metres on each segment. I think the reader should be told how this ditribution of slip was chosen? If various combinations of slip were tried in order to obtain good results, the reader should know this, or if the choice was made on a geological/geophysical basis we should also be told. As it stands the authors could be accused of biasing the results towards the hypothesis that they favour, since they did not also experiment with alternative slip models for hypothesis B.

**Answer 1:** We agree with the referee and added some new information in the manuscript. Firstly, we tested thrust and strike-slip sources with uniform slip to conclude that our preferred solution should be a thrust. This solution is compatible with the geodynamic setting. Later, to ameliorate the fit of the tsunami period, we tested the use of 4 segment sources with 50 km length each. We set the slip values to maintain a mean slip of 11 m. The location of the maximum slip area within the fault plan can hardly be deduced for non-instrumental events. Nevertheless, we tuned the slip distribution through the comparison of three setups: (1) maximum slip towards the SW, (2) maximum slip towards the NE and (3) maximum slip close to the center of the fault. In the first setup the withdraw historical observed at Cadiz is less evident and produces little inundation in Lisbon; in the second setup there is inundation at Cadiz, which is not supported by historical data. All these results led us to select the maximum slip at the center of the fault.

We also tested similar setups for the slip distribution for hypothesis B: 1) maximum slip towards the SW, (2) maximum slip towards the NE and (3) maximum slip close to the center of the fault. All these setups produce a very small withdraw in Cadiz (smaller than 0.3 m) which would be difficult to be observed. All the tested slip distributions for hypothesis B produced extreme wave heights (above 10 m) along the coast of Morocco. And these sources produce results that are less consistent with observations in comparison to hypothesis A in northern Europe.

Also, we checked a considerable number of finite source rupture models for several inverse earthquake mechanism (c.f. Yamanaka and Kikuchi (2003), Gusman et al. (2010), Wei (2014), Okuwaki et al. (2016)) suggests slip distributions with maximum values in the approximate center of the faults. However, the segmentation of the fault does not lead to a significant reduction of the wave period in Lisbon.

However, we included more information in the manuscript.

The manuscript in section 4.2, paragraphs 3-5 now read:

"**Hypotheses A and A-MS:** Here we use a strike angle compatible with the study by Martínez-Loriente et al., (2013) that follows the morphology of the Coral Patch scarp and seamount (Fig.

1 and Fig. 3). To take into account the tectonic regime of the source area we choose fault plane parameters compatible with a structure of compressive nature. The velocity vector predicted by NUVEL 1A (Fig. 3) together with the short tsunami wave periods (4-12 minutes) reported in 1761 (table 1) are in line with the chosen dip angle of 40 degrees (table 2). On the other hand, Martínez-Loriente et al. (2013) suggest for the Coral Patch Faults dip angles of 30±5 degrees dip and a rake angle of 90 degrees. These authors also conclude that the fault root is between 7 and 13 km depth. We approximate the rake angle according to the difference between the convergence arrow given by the circle around the Euler Pole and the fault plane (Fig. 3).

The wave period in Lisbon produced by this candidate source is 30 minutes. This value is not compatible with the observations (Table 1). Trying to solve this problem, we implemented a multi-segment fault here called A-MS. This multi-segment solution consists of four segments each 50 km. The four segments are placed adjacent to each other, and the rupture mechanism is equal for each segment as in hypothesis A with a mean slip of 11 m (Table 2). To investigate the slip distribution, we tested three setups: (1) maximum slip towards the SW, (2) maximum slip towards the NE and (3) maximum slip close to the centre of the fault. In the first setup the withdraw historical observed at Cadiz is less evident and produces little inundation in Lisbon; in the second setup there is inundation at Cadiz, which is not supported by historical data. All these results led us to select the maximum slip at the centre of the fault. The slip of each segment is presented in table 2. The synthetic waveforms are presented in figure 5 and discussed in sections 5 and 6.

**Hypothesis B:** Finally, we test an alternative hypothesis here called B with a larger strike-slip component compared to hypothesis A. This also results in larger fault length and a steeper dip angle. Here, we consider a rupture along a fault plane rotated about 180° when compared to hypothesis A. To do this, we selected compatible strike and rake angles that result in a sinistral inverse lateral rupture (table 2). The implementation of the different setups of slip distribution in solution B does not improve the quality of the results, therefore we only consider a single segment fault for this hypothesis. The synthetic waveforms are presented in figure 7 and discussed in sections 5 and 6."

**Comment 2:** Tsunami heights at a coastline can vary considerably depending on fine-scale bathymetric features close to the coast, so the application of Green's Law using an offshore coarse-grid wave amplitude to make comparisons with historical observations introduces a lot of uncertainty. I would like to see this better acknowledged. I think the reader should also be told how 5m was chosen for the depth at the shore used in Green's Law, since this seems rather arbitrary, and a different choice could lead to different conclusions.

**Answer 2:** We are aware of the limitations of the extrapolation methods to compute water elevations near the shore. However, the use of Green's law to amplify the maximum water-surface elevation can provide a first order approximation of the tsunami heights near the coast. This method was recently used by several authors (Davies et al. (2017); Brizuela et al (2014), Hebert & Schindelé (2015)) to overcome absence of high-resolution Digital elevation models. Additionally, Hebert and Schindelé (2015) conclude that extrapolation for depths between 10 and 1 m generally allows for a good fit with the observations. Our observations dataset contains incertitude inherent to historical observations. The selected extrapolation depth must be sufficiently close to the shore to be estimated by the eyewitnesses. To clarify this, we changed the manuscript accordingly.

The manuscript in section 4.1, paragraph 5 now reads:

"For the locations in Ireland, the United Kingdom, the Azores, Madeira and Barbados, we estimate the wave heights near the shore using the Green's Law (Green, 1838), following Hebert and Schindelé (2015) and Davies et al. (2017). Hebert and Schindelé (2015) concluded that the extrapolation for depths between 10 and 1 m generally allowed for a good fit with the observations for the 2004 Indian ocean tsunami. The Green's Law is based on the linear shallow water wave equations and allows to quickly approximate the amplification of wave heights at a shallower depth close to the shore when considering a plane beach. The wave height increases to the fourth root of the ratio between the depth at the shore and the water depth at the VTG. We extrapolate the maximum wave height values between the depths of the VTG (table 3 and 4) to points located at 5 m depth.

$$h_s = \sqrt[4]{\frac{d_s}{d_d}} * h_d \qquad \text{Eq. (1)}$$

Where $h_s$ and $h_d$ are the wave heights at the shore and the VTG respectively, and $d_s$ and $d_d$ are the depths at the shore and the VTG respectively. We use a constant value of 5 m which is sufficiently close to the shore to be observed by eyewitnesses. The results of the approximation according to the Green's Law are presented in table 3 and 4."

**Other points:**

**Comment 3:** There are a few minor grammatical errors, that should be picked up in copy-editing, so I will not list these here.

**Answer 3:** We used a spelling and grammar correction program and corrected all identified mistakes.

**Comment 4:** Figure 1 might be better presented as two full-width figures (maybe Figure 1a and 1b?). The problem is that the inset in Figure 1 is currently too small and hard to read.

**Answer 4:** We agree with the referee and produced figure 1 a.) and 1 b.)

**Comment 5:** The analysis of the combined effects of tide and tsunami amplitude in the discussion is a bit weak, as the maximum water level does not necessarily occur at the time of maximum tsunami amplitude (a different comination of tide + tsunami could be higher). I doubt this will affect the final conclusions (and would not insist that the authors do this), but in an ideal paper it would be better to show the modelled tsunami signal superimposed on top of a simple model for the tide.

**Answer 5:** We agree with the referee. We estimate the tide level from astronomical tables to conclude that the tide level is similar to the tide on the 4[th] of March 2016. We have used this data to plot the tsunami signals above the tidal signal for the 4[th] of March 2016. The results however do not change our conclusion. We leave the decision to the editor if it is necessary to include the figures.

[Figure]

**Figure 8. The comparison of the tsunami signal on mean sea level (msl) which is 2 m above Hydrografic Zero (HZ) for hypothesis A-MS compared with the tide signal similar for March, 31 1761.**

[Figure]

**Figure 8. The comparison of the tsunami signal on mean sea level (msl) which is 2 m above Hydrografic Zero (HZ) for hypothesis B compared with the tide signal similar for March, 31 1761.**

**Minor points:**

p1, line 8, replace 'set' with 'setting'

p1, line 15, should say '.. 1st November 1755 earthquake ...'

p2, line 4, if the inset to figure 1 was expanded it might be nice to identify the HAP on it.

p9, line 15, replace 'ancient' with 'historical'

p19, line 20, '... source A is ...' should probably be '... source A-MS is ...'?

**Answer Minor points:** We considered all the suggestions of the referee.

Yamanaka, Y., and M. Kikuchi. (2003). Source process of the recurrent Tokachi-oki earthquake on September 26, 2003, inferred from teleseismic body waves. Earth Planets and Space 55 (12):E21-E24.

[revised manuscript text omitted]